# Increased antioxidative defense and reduced advanced glycation end-product formation by metabolic adaptation in non-small-cell-lung-cancer patients

Tamara Tomin[1], Sophie Elisabeth Honeder [1], Laura Liesinger[1], Daniela Gremel[1], Bernhard Retzl[1], Joerg Lindenmann[2], Luka Brcic[3], Matthias Schittmayer [1] ✉ & Ruth Birner-Gruenberger [1,4] ✉

Reactive oxygen species can oxidatively modify enzymes to reroute metabolism according to tumor needs, rendering identification of oxidized proteins important for understanding neoplastic survival mechanisms. Thiol groups are most susceptible to oxidative modifications but challenging to analyze in clinical settings. We here describe the protein and small-molecular thiol oxidation landscape of 70 human lung tumors (and their paired healthy counter parts) and demonstrate that cancer adapts metabolism to increase glutathione synthesis to counteract oxidative stress. Glyoxalases, the key enzymes in the detoxification of methylglyoxal, a byproduct of glycolysis and precursor of advanced glycation end-products, are compromised by oxidation and down-regulation. Despite decreased methylglyoxal detoxification capacity, cancers do not accumulate advanced glycation end-products. Since in vitro down-regulation or inhibition of GAPDH upregulates glyoxalases, we propose that tumors reduce methylglyoxal by activating GAPDH.

Lung cancer is one of the most commonly diagnosed cancers worldwide and remains the leading cause of cancer-related deaths. Although lung tissue has evolved to withstand the increased oxidative exposure caused by its native function of oxygen uptake, additional oxidative stress plays a critical role in carcinogenesis of the lung[1,2]. Prolonged exposure to tobacco smoke, air pollution, pathogens and/or other risk factors submits lungs to a pro-oxidative environment, correlates with a functional decline of the lung epithelium[2–4] and can actively contribute to neoplastic transformation. On a chemical level, the negative effects of oxidative stress can be traced to the impact of reactive oxygen species (ROS), a group of unstable, electron-rich molecules that drive the oxidation of diverse biomolecules. While a small amount of ROS is common and even needed for physiological signaling[4], an excess of ROS can lead to permanent cell damage and cell death.

In cancer, ROS act as a double-edged sword: while some cancer therapies are based on triggering intracellular ROS to the point of cellular destruction[5], ROS-derived genetic instability and protein oxidation can also work in favor of pro-oncogenic signaling by modulating metabolic pathways and supporting neoplastic transformation[5–7]. One possibility for how crosstalk between ROS production and metabolism is achieved is via oxidative post-translational modifications (oxPTMs), especially on cysteine residues. Cysteine oxPTMs can lead to both loss[8] and gain of enzymatic functions[9], and correspondingly rewire cancer metabolism[10,11].

To obtain a deeper understanding of the crosstalk between oxidative stress, redox signaling and metabolism in lung cancer, we

[1]Institute of Chemical Technologies and Analytics, TU Wien, Vienna, Austria. [2]Division of Thoracic and Hyperbaric Surgery, Medical University of Graz, Graz, Austria. [3]Department of Pathology, Medical University of Vienna, Vienna, Austria. [4]Diagnostic and Research institute of Pathology, Medical University of Graz, Graz, Austria. ✉e-mail: matthias.schittmayer@tuwien.ac.at; ruth.birner-gruenberger@tuwien.ac.at

collected tumor and matched healthy tissue from 70 individuals with non-small cell lung carcinoma (NSCLC). NSCLC accounts for 84% of diagnosed lung cancers and is almost exclusively treated by surgical resection in early stages[12], providing access to often chemotherapy-naïve tissue specimens. Considering the importance of cysteines in redox signaling[13,14], our specific focus was the analysis of cysteine oxidation, which is inherently a challenging task. Thiol residues are extremely prone to artificial post-sampling oxidation, raising the need for special precautions during sample collection, including instant thiol quenching[15]. Thus, tissue samples (from both tumor and healthy sections of the incised lung piece) were preserved from post-sampling oxidation immediately upon collection in neutrally buffered 80% methanol containing the cysteine alkylating reagent N-ethylmaleimide (NEM) following our recently published method[15], enabling an unbiased determination of both small molecular weight (e.g., glutathione) as well as protein thiols.

By applying this approach, we demonstrate that lung tumors cope with rising intracellular oxidative stress levels by increasing glutathione biosynthesis as well as by modulating the abundance and redox state of a number of key players of cellular metabolism, especially related to glucose utilization. Interestingly, we observed an increase in oxidative glucose metabolism on a protein abundance level in tumor tissue, which is in line with previous proteomics studies[16,17]. The expression of many mitochondrial proteins (including those of the citrate cycle) was increased in the tumor, suggesting active mitochondrial function. We discovered a potentially redox-dependent hampering of the glyoxalase system in cancer, the main detoxifying route of methylglyoxal (MG), a major reactive side-product of glycolysis. In this regard, we identified glyceraldehyde-3-phosphate dehydrogenase (GAPDH) as the main regulator of the glyoxalase system in lung cancer cells. Our combined comprehensive ex vivo redox analysis suggests that lung cancer at the same time reinforces oxidative metabolism while reducing MG production by increased GAPDH activity. GAPDH, which is supposed to prevent accumulation of MG, was also found to be a target of MG modification[18], highlighting the interaction of oxidative stress and cellular metabolism.

## Results

### Intra- and extracellular changes in redox-balance in lung cancer

To address the impact of oxidative stress on tissue redox signaling in cancer we carried out redox proteomics profiling of 70 matched healthy and lung tumor tissue pieces which, upon site annotation and after filtering for valid values (see Materials and methods), enabled us to quantitatively address the oxidative status of 1834 individual cysteine residues across 921 different proteins (Supplementary Data 3). The differences in redox ratios between the tumor and healthy tissues were not substantial; besides a couple of outliers, the samples did not strongly separate in a principal component analysis (Suppl. Fig. S2A) and the median $Cys_{red}/Cys_{ox}$ ratio of all samples was always clustered around the value of 1 (Suppl. Fig. S2B), suggesting that (a) redox ratios are biologically tightly controlled and (b) our instantaneous alkylation protocol yielded unbiased results. While 97% of all proteins contain cysteine, reportedly only 18% of all proteins are expected to harbor disulfide bonds[19] other thiols are then free or otherwise modified.

Of the 1834 confidently quantified cysteine redox ratios, we found 170 cysteine residues to be differentially oxidized upon multi-testing corrected student t-testing (Fig. 1A, marked in black) and 488 residues upon analysis with a less stringent significance threshold (no multi-testing correction; Supplementary Data 3). The two top-most significantly changed oxidized cysteine residues in tumors were $Cys^{156}$ of caveolin-1 (CAV1) and $Cys^{240}$ of receptor of activated protein C kinase 1 (RACK1) (Fig. 1A, Supplementary Data 3). These are particularly intriguing findings as both proteins are direct interactors of potent oncogenes. RACK1 interacts with and stabilizes activated protein kinase C

and binds and inhibits SRC[20]. $Cys^{156}$ of CAV1 is a known palmitoylation site[21] as well as a nitrosylation target[22]. S-palmitoylation, in contrast to S-nitrosylation, is not reduced by TCEP and thus, in contrast to the latter, not detected by our approach. When modified by S-nitrosylation bound proto-oncogene kinase SRC is displaced and activated, thereby phosphorylating CAV1 on Tyr-14, which leads to destabilization of CAV1 oligomers[23]. It is additionally noteworthy to mention that CAV1 gene expression is often reduced in lung cancer compared to healthy parenchyma, and we observe the same on protein level[24] (Supplementary Data 2). On the other hand, among the most oxidized proteins in healthy tissue compared to tumor were different isoforms of myosin, including MYH8 and MYH11 (Fig. 1A; Supplementary Data 3). Next to increased oxidation, there was a greater abundance of myosin in healthy tissue on protein level, of both conventional as well as non-muscle isoforms (Supplementary Data 2); with exception of MYO1E, a myosin we detected to be more abundant in tumor (Supplementary Data 2) and which was recently proposed as an independent marker of mortality in lung cancer[25].

To obtain a better overview of oxidative stress-affected cellular compartments and gain a deeper understanding of the crosstalk between redox signaling and metabolism, the 488 significant site hits from redox profiling were analyzed for the subcellular localization of their corresponding proteins and further subjected to gene ontology enrichment of biological processes (GOBP). For the localization analysis, we first inspected the overlap of oxidation-affected proteins between tumor and healthy tissue: 33 proteins were both more oxidized and more reduced in tumors, while 150 and 186 were uniquely more oxidized in tumor or healthy tissue, respectively (Fig. 1B; Supplementary Data 4). Of those 150 more oxidized proteins in tumors, the large majority (63.33%) were cytosolic; 18.67% organelle/membrane bound, and only 10.67% extracellular/secreted. From the 186 uniquely more oxidized proteins in healthy tissue, the contribution of cytosolic proteins was lower (42.62%) while the percentage of oxidation affected extracellular proteins more than doubled compared to tumor tissue (24.73% vs. 10.67%). Interestingly, this was also reflected in the protein abundance level of antioxidative enzymes. For example, while cytosolic and mitochondrial superoxide dismutase (SOD1 and SOD2) were more abundant in tumor tissue (especially the mitochondrial isoform SOD2), SOD3, the excreted isoform, was significantly more abundant in healthy tissue (Fig. 1C; Supplementary Data 2). Blood-cell-related oxidative enzymes such as MPO and eosinophil peroxidase (EPX) were also significantly more abundant in healthy tissue (Fig. 1C and Supplementary Data 2). In general, antioxidative proteins were more abundant in healthy compared to tumor tissue (Supp. Fig. S1C, marked in pink; Supplementary Data 2). A general overview of changes in protein abundances between healthy and tumor tissue is presented in Suppl. Fig. S1 as upregulated and downregulated GOBP with significantly affected proteins as input for enrichment.

In summary, our data suggest that relative to healthy tissue, tumors experience more intracellular oxidative stress, triggering intracellular antioxidative support. On the contrary, healthy tissue displayed higher levels of extracellular oxidative stress, which could be a consequence of better vascularization and therefore oxygen availability in healthy lung tissue: while the reported oxygen concentration for healthy lung tissue is -5.6%, for lung tumors this value drops to -2%, and in some extreme cases even to as low as 0.1%[26,27]. Along these lines, our quantitative proteomic analysis of protein abundance revealed a prominent reduction in abundance of all subunits of hemoglobin in tumor tissue (Supplementary Data 2, 5) as well as a reduction in proteins responsible for maintaining proper lung function and integrity of the epithelial barrier, such as claudins[28] (claudin-18), caveolins

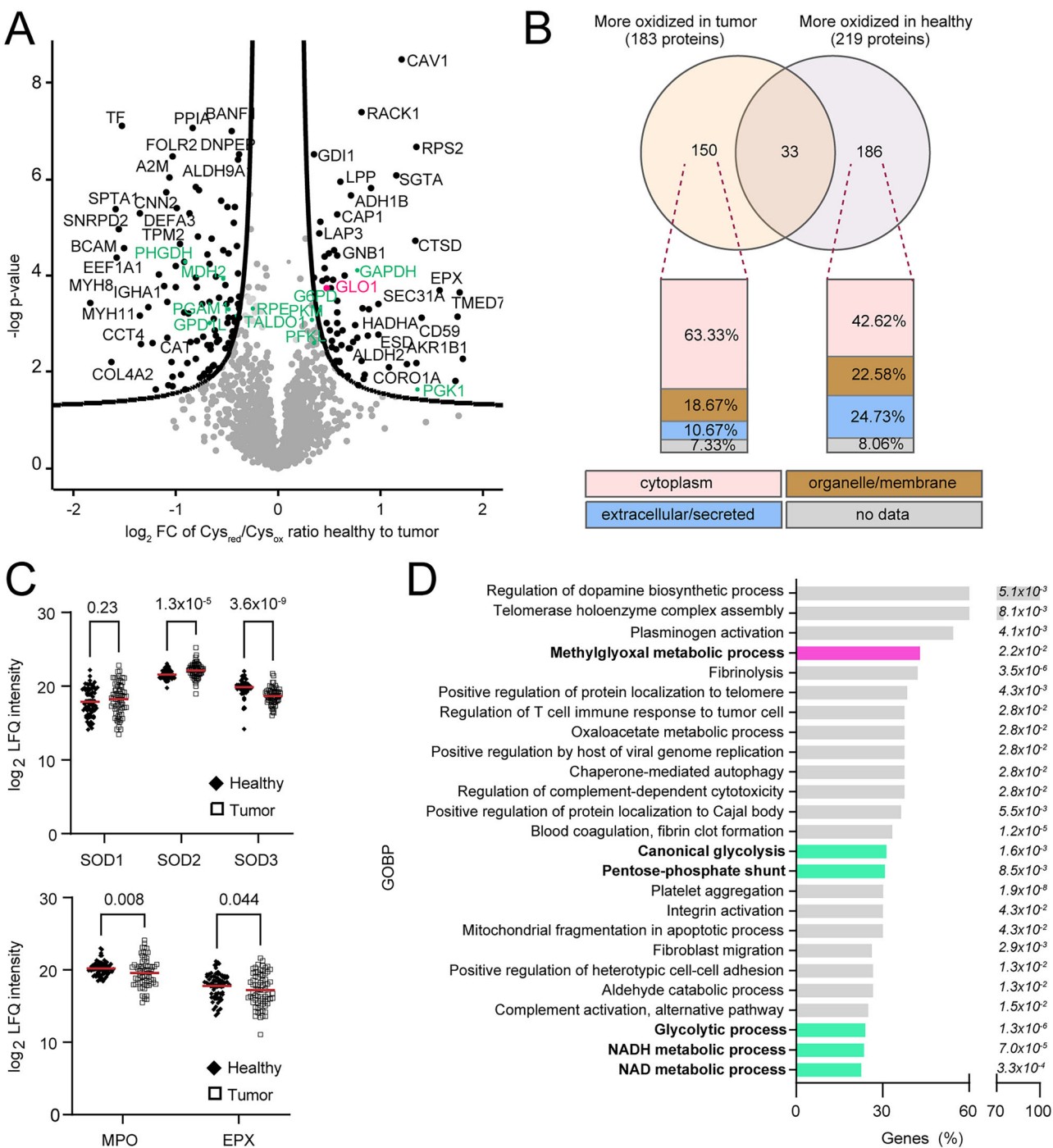

**Fig. 1 | Redox proteomics of lung tumor and tumor-adjacent healthy tissue reveals intra- and extracellular differences. A** Volcano plot depicting proteins whose redox state was significantly affected in tumor versus healthy tissue. Each dot represents an individual cysteine residue, i.e., median value of L/H ratios of all peptides harboring a given cysteine (labeled black: significantly changed proteins (FDR corrected $p$ value < 0.05, S0 = 0.1); specific pathways of interest, e.g., glucose and methylglyoxal metabolism, are highlighted in green and pink, respectively). **B** Venn diagram depicting a number of overlapping and significantly more oxidized proteins in tumor vs. healthy tissue. According to the UniProt annotation of individual proteins for their subcellular location, tumor tissue showed higher oxidation of intracellular proteins and lower oxidation of extracellular proteins than healthy tissue. **C** (top) Protein abundances (LFQ values) of superoxide dismutase (SOD) 1 and 2 (cytosolic and mitochondrial isoform; $N$ = 64 (SOD1) and 69 (SOD2) paired

tumor and healthy tissue pieces, two-sided paired $t$ tests) were higher while protein abundance of SOD3 (extracellular, secreted isoform; $N$ = 66 two-sided paired $t$ tests) was lower in tumor tissue; (bottom) Two extracellular oxidative proteins, namely myeloperoxidase (MPO) and eosinophil peroxidase (EPX) were also lower in tumor tissue ($N$ = 69 (MPO) or 68 (EPX) paired tumor and healthy tissue pieces; two-sided paired $t$ tests). **D** The top 25 non-redundant significantly enriched terms upon GOBP enrichment analysis with all significantly (Student's $t$ test $p$ value < 0.05) redox-affected proteins as input (FDR for enrichments was always <0.05). Genes (%) represents the number of matched genes to the total size of the term. Exact FDR values for enrichment are noted on the side of the bar chart. Supplementary Data 3 and 4 are source data for **A**, **B** and **D**, while for **C** source data is provided in Source Data File.

(CAV1 and 2)[29], angiotensin-converting enzyme (ACE1 and 2)[30] and others (Suppl. Fig. S2C, A; Supplementary Data 2, 5).

## Crosstalk between redox signaling and glucose metabolism in lung tumor tissue

To answer whether higher intracellular oxidative stress and disturbed redox signaling affect metabolic enzymes of cancer cells, we also carried out a GOBP enrichment with the 369 significantly redox-affected proteins (matching 488 significantly affected Cys residues) as input. This approach revealed prominent changes in the redox state of enzymes involved in glucose metabolism, especially in glycolysis and in the pentose-phosphate pathway (PPP; Fig. 1D, marked in green), as well as of enzymes involved in detoxification of detrimental side products of glycolysis (i.e., methylglyoxal (MG); Fig. 1D (marked in pink) and Supplementary Data 4). Given the intertwining nature of these pathways, we merged our redox (Fig. 1A and Supplementary Data 3) and label-free quantitative proteomics data (Supplementary Data 2 and Suppl. Fig. S2C) and depicted combined effects on enzymes involved in glucose metabolism in tumor (as compared to healthy tissue; Fig. 2).

This analysis revealed that not only were the protein levels of glycolytic enzymes changed in lung cancer, but also their redox states. It is well known that cancer cells thrive on glucose metabolism and are also able to shape it to their needs by changing protein expression and allosteric regulation of the enzymes involved[31]. In this large patient cohort, we provide patient-derived in vivo evidence that a number of glucose-related metabolic enzymes are also susceptible to oxPTMs, which might affect their enzymatic functions. We detected higher oxidized states of $Cys^{170}$ of phosphofructokinase (PFK), $Cys^{247}$ of glyceraldehyde-3-phosphate dehydrogenase (GAPDH), $Cys^{379/80}$ of phosphoglycerate kinase (PGK) and $Cys^{49}$ and $Cys^{326}$ of pyruvate kinase M (PKM), which were all reported to interfere with either the primary or secondary function of these enzymes[32–35]. Any impairment in the activity of glycolytic enzymes might trigger flux rerouting towards antioxidant production e.g., to the PPP for NADPH regeneration (and with it, glutathione recycling via glutathione reductase (GR)) or to serine-biosynthesis[36] (Fig. 2). Serine biosynthesis can drive the S-adenosyl methionine (SAM) cycle via one carbon metabolism, thereby fueling the trans-sulfuration pathway and ultimately increasing glutathione production.

In line with the latter, we detected an increase in protein levels of glutathione synthase (GSS), the enzyme responsible for addition of glycine to the γ-glutamyl-cysteine in GSH biosynthesis (Figs. 2 and 3A) and prominent accumulation of glutathione, both in its reduced and oxidized form (Fig. 3B) as well as enzymes involved in serine de novo synthesis[36] (Fig. 3A). While glutathione accumulation has been already reported in NSCLC[37], we further demonstrate that not only GSH but also its precursors, γ-glutamyl-cysteine (GluCys), cysteine and homocysteine (HCys) are increased in tumor tissue, as well as the degradation product of glutathione, cysteinyl-glycine (CysGly; Fig. 3C), in line with the observed greater abundance of glutathione hydrolase (GGT) in cancer tissue (Fig. 3A; Supplementary Data 2). GGT is located on the extracellular surface with the catalytic site oriented to the extracellular environment[38], thus scavenging extracellular glutathione. Overall, it seems that even when the tumor is exposed to increased oxidative stress (as suggested by the increase in glutathione disulfide (GSSG) levels), it is still able to foster its antioxidative defense and protect itself.

It is, however, noteworthy to mention that for a number of glycolytic enzymes, we did not find the reported "most critical" cysteine residue (i.e., the active-site cysteine or key cysteine for enzymatic function) to be more oxidized in tumors. For example, while we detected higher oxidation of $Cys^{49}$ and $Cys^{326}$ of PKM, the oxidative state of $Cys^{358}$, a cysteine residue which when oxidized leads to complete loss of PKM2 activity prompting metabolic flux routing towards

PPP[8] was not significantly changed (Fig. 2). The same is the case for the catalytic $Cys^{152}$ of GAPDH[39,40] (Fig. 2). Modifications of cysteines even distal from the active site might however impact enzymatic activity, and for both PKM2 as well as for GAPDH, protein levels were prominently increased in tumor tissue (Fig. 2; Supplementary Data 2).

Interestingly, the abundance of the enzymes whose key cysteine was observed to be less oxidized in tumor (marked in green in Fig. 2) remained unchanged, as was the case for phosphoglycerate mutase 1 (PGAM1), transaldolase 1 (TALDO1) and 6-phosphoglucolactonase (PGLS). Only aldolase A (ALDOA) was found to be both less oxidized on its potentially critical cysteine ($Cys^{339}$) and at the same time to be more abundant, which may corroborate its critical role in cancer survival and metastasis[41].

However, not only was the abundance of enzymes involved in glycolysis affected, but also those acting in the tricarboxylic acid (TCA) cycle. While the redox status of TCA enzymes was mainly unchanged (except for malate dehydrogenase at $Cys^{89}$ and α-ketoglutarate dehydrogenase at $Cys^{802}$), almost all TCA enzymes were more abundant in tumor cells. This was also reflected in the GOBP enrichment of all more abundant tumor proteins as input: a prominent cluster of mitochondrial activity/oxidative respiration GOBP terms emerged as upregulated in tumor tissue (Suppl. Fig. S1B; Supplementary Data 6). A similar pattern of TCA enzyme abundance in tumor vs. healthy tissue was also reported in two other, independent NSCLC studies[16,17] (Suppl. Fig. S2D). In summary, our analysis corroborates earlier in vivo tracing studies that lung cancer cells heavily rely on the TCA cycle for energy production[42]. Attempting to maintain mitochondrial functionality amid the lack of oxygen might be a significant contributor to the rising pool of ROS[43]. Mitochondrial resilience in response to nutrient stress was recently also shown in vitro in NSCLC cell models under conditions of cysteine starvation, revealing that sustaining the mitochondrial cysteine pool by GSH catabolism can support Fe-S proteins and thus mitochondrial respiratory function[44]. As aforementioned, we observe accumulation of both GSH (Fig. 3B) and its degradation product CysGly (Fig. 3C), as well as SOD2 (the mitochondrial isoform of superoxide dismutase) in tumor tissue (Fig. 1C), suggesting that tumor mitochondria are in greater need of antioxidative protection.

Lastly, it is also noteworthy that two enzymes of the non-oxidative branch of the PPP were found less oxidized in cancer (Fig. 2), and protein levels of 6-phosphogluconate dehydrogenase (6PGD) were increased (Fig. 2; Supplementary Data 2). 6PGD is one of the two critical enzymes for NADPH regeneration, and with it, for overall intracellular redox balance[45,46].

## Redox and protein level downregulation of glyoxalase system

Next to glycolysis one of the top enriched GOBP terms from our lung cancer redox-proteomics dataset was methylglyoxal metabolism (Fig. 1D, marked in pink). Methylglyoxal (MG) is a highly reactive and toxic side-product of glucose, lipid and protein metabolism, with glycolysis being its major source[37,47,48]. MG is mainly detoxified by a two-enzyme system: in the first step, glyoxalase 1 (GLO1) with the help of a molecule of GSH creates the intermediate product S-lactoylglutathione (sLG)—a metabolite consequently hydrolyzed by the activity of hydroxyacylglutathione hydrolase (GLO2; HAGH), which produces D-lactate and recycles the used molecule of GSH (Figs. 2 and 4A). Resulting D-lactate can be consequently converted to pyruvate with the help of lactate dehydrogenase D[49] (LDHD; Fig. 2).

In our dataset, we detected prominent oxidation of $Cys^{139}$ of GLO1 in lung tumors (Figs. 1A, 2 and 4A, Supplementary Data 3), a cysteine residue deemed critical for the enzyme's activity in vitro[50]. To validate if some degree of GLO1 inhibition is present in tumor tissue, we assessed levels of sLG, the direct GLO1 product, and accordingly observed a trend of reduced levels of sLG in tumors ($N = 70$; Fig. 4A), further indicating potentially lower activity/lower flux through the GLO1 enzyme in our patient cohort. Interestingly, contrary to $Cys^{139}$ of

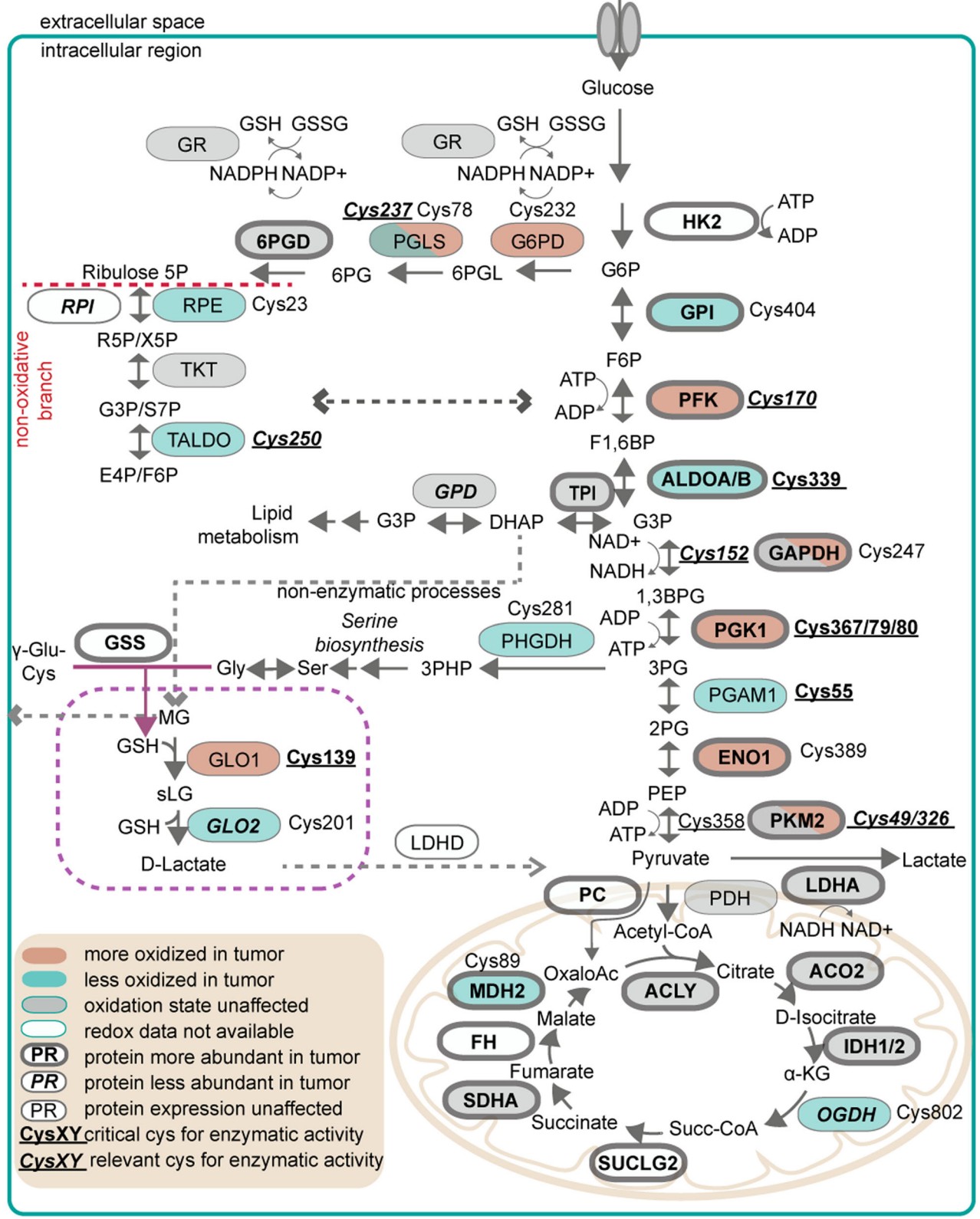

GLO1, Cys[201] of GLO2 was significantly less oxidized in tumor tissue (Figs. 2 and 4A; Supplementary Data 3). While Cys[201] of GLO2 was identified as an S-nitrosylation target[51], nothing is known with regard to its effect on enzyme activity.

Overall, the more oxidized state of GLO1 together with lower sLG levels suggests a potentially reduced ability of tumor cells to metabolize MG or perhaps less need for the cells to metabolize it.

Although GLO1 is reported to be higher expressed in a number of different cancer types including the lung[37,52,53], in our cohort of 70 patients we did not observe such a clear trend: almost half of the patients had a higher and the other half a lower GLO1 protein expression in tumor compared to the surrounding healthy tissue (Fig. 4A; Supplementary Data 2). On the other hand, GLO2 expression was consistently reduced in tumor (Fig. 4B;

**Fig. 2 | Oxidative and abundance changes of glucose and methylglyoxal metabolic enzymes in lung cancer patients.** Glycolytic enzymes were not only more abundant (marked with thick margin and bold writing) but also often more oxidized (red colored) in lung tumors. Oxidation might interfere with enzyme's functionality. Main methylglyoxal (MG) detoxifying enzymes glyoxalase 1 (GLO1) and glyoxalase 2 (GLO2) seem to be less active (GLO1) and less abundant in tumor (GLO2). Furthermore, almost all enzymes of the citrate cycle were upregulated in tumor. Significance threshold for both oxidation and abundance: p value < 0.05 (unpaired two-sided Student's t test). 1,3BPG 1,3-bisphosphoglycerate, 2PG 2-phosphoglycerate, 3PG 3-phosphoglycerate, 3PHP 3-phosphohydroxypyruvate, 6PG 6-phosphogluconate, 6PGD 6-phosphogluconate dehydrogenase, 6PGL 6-phosphogluconolactone, ACO2 aconitase, ALDOA/B aldolase A/B, ACLY citrate synthase, αKG α-ketoglutarate, DHAP dihydroxyacetone phosphate, E4P/F6P erythrose 4-phosphate or fructose-6-phosphate, ENO1 enolase 1, F1,6BP fructose 1,6-bisphosphate, FH fumarase, G3P glycerol-3-phosphate, G3P/S7P G3P or sedoheptulose 7-phosphate, G6P glucose-6-phosphate, GAPDH glyceraldehyde-3-phosphate dehydrogenase, GPD glycerol-3-phosphate dehydrogenase, G6PD glucose-6-phosphate dehydrogenase, GPI glucose-6-phosphate isomerase, GR glutathione reductase, GSH glutathione, GSS glutathione synthetase, GSSG glutathione disulfide, HK2 hexokinase 2, IDH1/2 isocitrate dehydrogenase 1/2, LDHA lactate dehydrogenase A, LDHD probable D-lactate dehydrogenase, MDH2 malate dehydrogenase 2, OGDH oxoglutarate (α-ketoglutarate) dehydrogenase, OxaloAc oxaloacetate, PC pyruvate carboxylase, PDH pyruvate dehydrogenase, PEP phosphoenolpyruvate, PFK phosphofructokinase, PGAM1 phosphoglycerate mutase 1, PGK1 phosphoglycerate kinase 1, PGLS 6-phosphogluconolactonase, PHGDH phosphoglycerate dehydrogenase, PKM2 pyruvate kinase M2, R5P/X5P ribose or xylulose 5-phosphate, RPE ribulose-phosphate-3-epimerase, RPI ribose 5-phosphate isomerase, SDHA succinate dehydrogenase, sLG S-lactoylglutathione, Succ-CoA succinyl CoA, SUCLG2 succinate-CoA ligase, TALDO transaldolase, TKT transketolase, TPI triosephosphate isomerase.

Supplementary Data 2). Interestingly, the more GLO1 was oxidized (the lower L/H of its Cys[139] was), the more GLO1 protein was produced, arguably as a compensation for the loss of enzymatic activity (Fig. 4B, left). The oxidation state of GLO1 did not correlate with GLO2 protein abundance (Fig. 4B, right). Of note, LDHD, the most downstream enzyme of the GLO pathway, was detected only in very few samples of our cohort in both tumor and healthy tissue, with no striking change in abundance (Fig. 2; Supplementary Data 2). Since almost nothing is known regarding GLO2 expression in lung cancer, we examined publicly available cancer genome atlas datasets (TCGA; source: Xena browser; TARGET GTEx dataset) for *GLO1*, *HAGH* (GLO2), and *LDHD* mRNA expression across normal, healthy lung and tumor-adjacent lung tissue as well as primary tumor of the lung. According to the TCGA public datasets, while *GLO1* mRNA expression was increased in both tumors and tumor-adjacent tissue compared to healthy lung (Fig. 4C), the opposite was the case for *HAGH* (GLO2) and *LDHD*: both *HAGH* (GLO2) and *LDHD* mRNA expression were prominently decreased in primary tumor compared to both normal and tumor-adjacent tissue (Fig. 4C). In all cases, however, tumor adjacent tissue demonstrated higher mRNA transcript levels of both GLO enzymes as well as *LDHD* compared to healthy lungs, indicating that the cells in the tumor microenvironment might have to cope with higher MG levels than usual. Low expression of *HAGH* (GLO2) as well as of *LDHD* was also prominently connected with poorer 5-year overall survival across different cancers (TCGA PANCAN dataset; N = 11,506 patients; Fig. 4D). Contrary to *HAGH* (GLO2) and *LDHD*, expression of *GLO1* had a minimal effect on survival. In addition to TCGA, we examined GLO expression in other published NSCLC proteomic studies[16,17] and observed similar effects as in our study: compared to healthy tissue, both GLO1 and GLO2 were prominently downregulated, while hypoxic targets were all upregulated in tumor (Fig. 4E). As one key property which distinguishes tumor to normal tissue is hypoxia, we wondered if hypoxia itself might play a role in inhibition of the glyoxalase pathway as hypoxia-driven GLO1 deactivation was suggested already by previous studies[54,55]. To that end, we exposed H358 lung cancer cells for 48 or 72 h to 1% oxygen and analyzed changes in their proteome and relevant thiols. Exposure to hypoxia (compared to the corresponding 21% control) led to an increase in sLG levels in H358 cells (Fig. 4F); and, on protein level, an inverse correlation between LDHA/HK2/GAPDH (only for HK2 the correlation was significant) and GLO2 (but not GLO1) abundance could be observed (Fig. 4G, H), suggesting that acute hypoxia can indeed influence the glyoxalase system, namely through downregulation of GLO2. In our patient cohort, however, an increase in LDHA was correlated with lower GLO1 but not GLO2 abundance (Suppl. Fig. S3).

## Downregulation of glyoxalase system does not result in MG accumulation

Lower activity of the glyoxylate system through ROS deactivation and/or hypoxia-driven downregulation would be expected to lead to local MG accumulation in tumor tissue, especially since solid tumors are known to exhibit a high level of glycolytic dependency[56]. As a result, the most common MG-driven protein modifications, namely MG-derived hydroimidazolone (MG-H1; +54.0106 Da) and dihydroxy MG/MG-derived carboxyethyl lysine (CEL; +72.0211)[57] should be increased. To this end, we ran an open search on the proteomics data of the collected tumor and healthy lung tissue and to our surprise, observed the opposite: tumors had fewer MG-driven modifications (namely MG-H1 and CEL) compared to healthy tissue (Fig. 5A and Supplementary Data 7). The result of the open search was further corroborated by a specific search targeting MG-H1 as variable modification on peptides (Fig. 5B and Supplementary Data 7), which confirmed a significantly higher abundance of MG-H1 modified peptides in healthy tissue compared to tumor across 70 patients (MG-H1 peptides more abundant in healthy are marked in violet). The observed lower MG metabolism and at the same time less MG presence might suggest that in our patient cohort, the glycolytic flux is efficiently pushed towards "lower" glycolysis, reducing the pool of available DHAP/G3P. The key enzyme to determine the flux, positioned at the crossroad between upper and lower glycolysis, is GAPDH (Fig. 5C). In our patient cohort higher GAPDH abundance was correlated with lower MG-H1 modification frequency, less sLG accumulation in tumor as well as lower GLO1 but not GLO2 levels (Fig. 5D). According to the TCGA database, high expression of *GAPDH* is also a predictor of poor survival across all cancer types (PANCAN dataset; N = 11,506 patients; Fig. 5E) and in our patient cohort, GAPDH abundance was directly correlated with tumor's glucose uptake (measured as fluorodeoxyglucose (FDG) and displayed as SUVmax values; Fig. 5F). Lastly, higher GAPDH abundance was also significantly correlated with more GLO1 oxidation, further strengthening the connection between high GAPDH presence and glyoxalase system inactivity (Fig. 5G).

## Crosstalk between GAPDH and the GLO system

To validate the influence of GAPDH on the GLO system we reduced GAPDH activity in vitro in the following manner: I) through siRNA-mediated knockdown of GAPDH; or II) pharmacological inhibition using koningic acid (KA) in A549 and H358 lung cancer cells. We hypothesized that limiting the flux through GAPDH would increase G3P and DHAP levels and thus cause MG formation, which would trigger upregulation of the GLO system for detoxification (Fig. 6A). Indeed, GAPDH knockdown

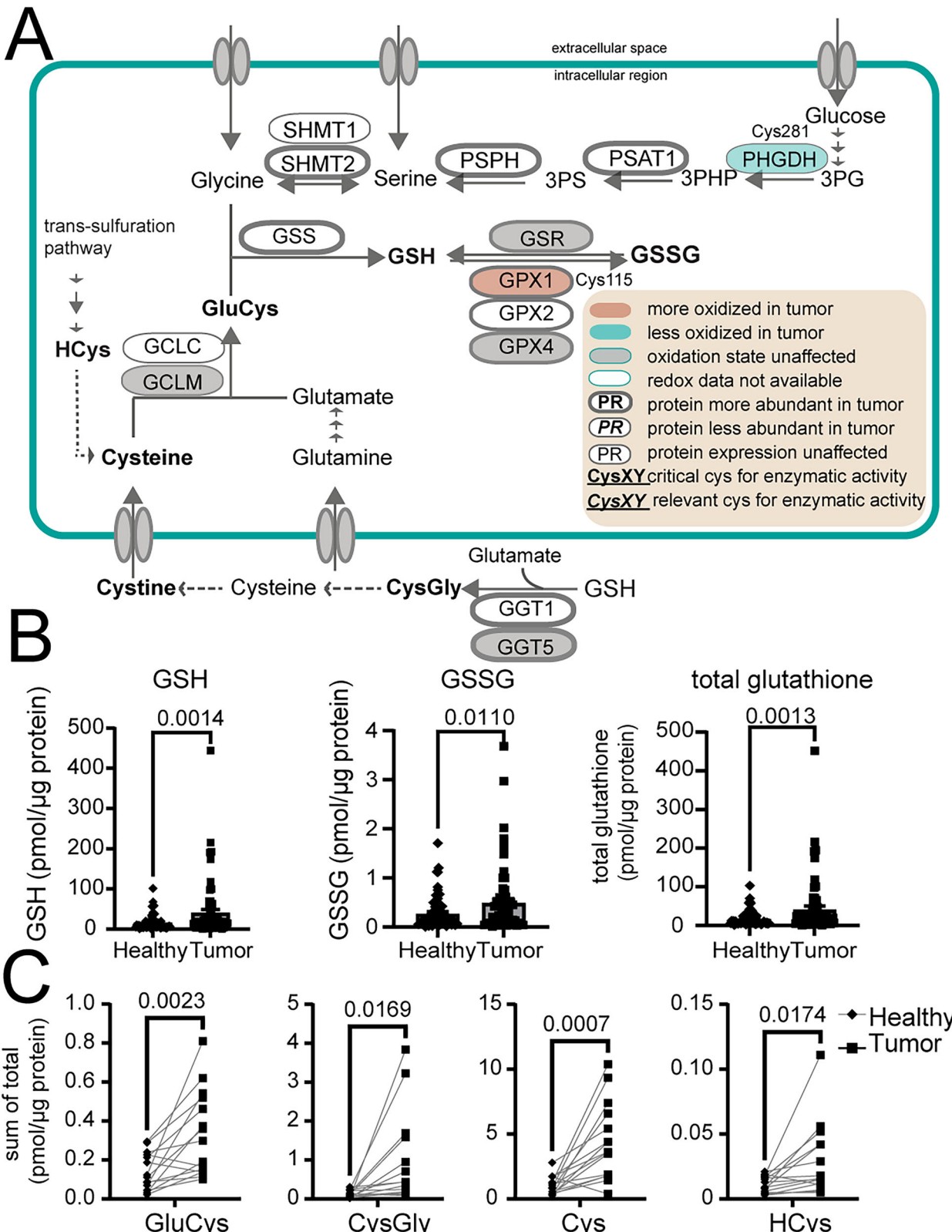

prominently increased GLO1 protein abundance in both cell lines, slightly driving TPI1 upregulation as well (Fig. 6B, C; Supplementary Data 8). Of note, for GLO1 the effect was more prominent in A549 cells with a more efficient knockdown.

Similar effects were observed for inhibition of GAPDH activity with KA, a potent and selective GAPDH inhibitor[58]. KA concentration was optimized for complete inhibition of GAPDH

activity (Fig. 6D) while cells were still proliferating after 24 h albeit at a reduced rate (Fig. 6E). GAPDH inhibition led to a prominent rise in both GLO1 and/or GLO2 in both cell lines (Fig. 6F; Supplementary Data 8 for both A549 and H358). In addition, both GAPDH KD and KA treatment increased the levels of sLG (Fig. 6G), further corroborating GAPDH as critical regulator of MG production and GLO system activity in lung cancer.

**Fig. 3 | Tumors accumulate glutathione and increase its biosynthesis.**
**A** Overview of serine/glycine and glutathione de novo synthesis. Most of the enzymes involved in the serine biosynthetic pathway were significantly upregulated in tumor tissue, together with glutathione synthase (GSS) and glutathione hydrolase 1/5 (GGT1/5); $N$ of tested paired tumor and healthy tissue for significantly changed enzymes (two-sided, non-corrected paired Student's $t$ test $p$ value < 0.05): 61 (PSAT), 44 (PSPH), 68 (SHMT2), 70 (GSS), 69 (GCLC), 51 (GGT1), 70 (GGT5)). **B** As a result, a prominent accumulation of both reduced (GSH) and oxidized glutathione (GSSG), and, thus, total glutathione (the sum of GSH and GSSG) could be observed within tumor tissue ($N$ of patients per group (healthy or tumor): 70, two-sided, paired Students' $t$ test; data represents mean ± standard error of mean

(SEM)). **C** Glutathione precursors and degradation products presented as sums of their oxidized and reduced forms ($N$ = 15 patients per group; two-sided paired Student's $t$ test). All thiol values are normalized to protein content of the tissue. Source data for **A** is Supplementary Data 2 and 3 and for **B**, **C** is the Source Data file. 3PG 3-phosphoglycerate, PHGDH phosphoglycerate dehydrogenase, 3PHP 3-phosphohydroxypyruvate, PSAT1 phosphoserine aminotransferase 1, PSPH phosphoserine phosphatase, SHMT1/2 serine hydroxymethyltransferase 1/2, GCLC/M glutamate-cysteine ligase catalytic/modifier subunit, GPX 1/2/4 glutathione peroxidase 1/2/4, GluCys γ-L-glutamyl-L-cysteine, CysGly cysteinyl glycine, HCys homocysteine.

## Discussion

ROS play an important role in development and progression of lung cancer orchestrating an interplay between oncogenes, oxPTM driven redox signaling and metabolic adaptations[59]. Tumorigenesis is a process driven by oncogenes, some of which are even known to enhance oxidative metabolism (e.g. *KRAS*), and with it, increase cellular stress exposure[60]. Tumors may employ targeted oxPTMs to both sustain cancer pro-oncogenic signaling and metastasis[20,22], but also to directly influence metabolic pathways. For example, to lessen their oxidative burden, cancer cells are able to use oxPTMs to reroute metabolic pathways according to their needs, e.g., for antioxidant production/recycling. Accordingly, inhibition of key glycolytic enzymes such as PKM2 and GAPDH, through oxPTMs (on Cys358 and Cys152, respectively), can divert glucose away from glycolysis towards PPP for regeneration of redox equivalents which can be forwarded to enzymes such as GR to enable glutathione recycling[8,61] (Fig. 2). While we did not observe a significantly higher oxidation of these two cysteine residues of PKM and GAPDH in lung tumors of our patient cohort (Fig. 2; Supplementary Data 3), we detected prominent oxidation of other cysteine residues of both GAPDH and PKM (Fig. 2; Supplementary Data 3), some of which, in case of PKM2, have been implicated with loss of enzymatic function[32]. However, we also see substantial elevation of protein abundance of both enzymes in cancer, which may compensate for the oxidative damage. Next to GAPDH and PKM2, other glycolytic enzymes such as PFK, PGK1 and enolase 1 (ENO1) were more oxidized but also higher abundant in tumors (Fig. 2; Suppl. Data 2,3). Oxidation of these enzymes has been reported to lead to a reduction of their activities[33,35,62], enabling diversion of carbon flux from glycolysis towards nucleotide and glutathione synthesis[62]: Slowing glycolysis down enables for glycolytic intermediates to be pushed towards serine synthesis and consequently folate cycle and/or trans-sulfuration pathway[63] to provide glycine and cysteine, the building blocks for glutathione production (Figs. 2 and 3A)[36]. In line with this notion, we also observe a prominent increase of glutathione synthase in tumors (GSS; Figs. 2 and 3A; Supplementary Data 2) as well as the serine de novo synthesis machinery (Fig. 3A, Supplementary Data 2), suggesting that indeed higher glutathione production takes place. Cancers, and especially lung cancer, are known to accumulate both reduced as well as oxidized glutathione[37,64]. We observed the same in our cohort of 70 patients: compared to patient matched healthy tissue, tumors had significantly higher levels of GSSG which they appeared to redox-balance by enhancing GSH production, reflected by a an accumulation of GSH as well as its precursors (Fig. 3B, C) and even the extracellular hydrolysis product (CysGly; Fig. 3C). In this regard, the thiol status of tumors aligns perfectly with enzyme abundances, as both GSH synthesis (e.g. GSS) as well as the extracellular GSH scavenging enzymes (GGT1/5) were prominently upregulated in tumor tissue (Fig. 3A; Supplementary Data 2). Overall, our snapshot of the redox-proteomic tumor landscape suggests that cancer cells endure higher levels of oxidative stress but are able to regulate glycolysis through redox-signaling to divert carbon flux towards antioxidant production, alleviating the oxidative stress burden.

Another reason for cancer cells to slow down glycolysis would be to reduce accumulation of toxic glycolytic intermediates, such as MG. Accumulation of advanced glycosylation end-products and dicarbonyl stress arising from uncontrolled MG reactivity can act as a double-edged sword in cancers - sometimes supporting tumor growth, while other times triggering detrimental inflammation and apoptosis[65,66]. In lung cancer, specifically, it has been shown that tumors tend to upregulate GLO1, the main enzyme for MG metabolic clearance, and accumulate sLG, the intermediate product of MG catabolism[37]. Accumulation of sLG could be either due to a higher activity/abundance of GLO1[37] or lower activity/abundance of the subsequent enzyme in the pathway - GLO2[67,68]. However, in our large cohort of 70 patients we observed the opposite: lung tumors exhibited lower sLG levels and fewer MG-derived modifications compared to healthy tissue (Figs. 4A and 5A, B). We thus hypothesize that less MG is produced in tumors by concomitant lower oxidation of DHAP/G3P to MG, due to a more efficient lower glycolysis with highly active serine de novo/glutathione synthesis and active TCA, reducing the pool of available DHAP/G3P for MG formation. In tumors we observe both, a rise in glutathione synthesis as well as an increase in the abundance of GAPDH and the whole TCA cycle enzymatic machinery (Fig. 2; Supplementary Data 2), in line with two independent NSCLC proteomics studies[16,17] ($N$ = 99–101, Fig. 4E; Suppl. Fig. S2D).

A key enzyme candidate to enable such versatility in routing of the glycolytic intermediates is GAPDH. High GAPDH activity in relation to the activities of the other glycolytic enzymes in the cascade can increase the pool of metabolites of lower glycolysis and funnel glycolytic intermediates towards both serine de novo synthesis and/or the TCA cycle depending on PKM activity, and at the same time reduce the pool of available DHAP/G3P needed for MG production. Indeed GAPDH was already recognized as a regulator of MG production in red blood cells of diabetic patients[69]. In our lung tumor patient cohort, rise in GAPDH abundance was positively correlated with glucose uptake of tumors and GLO1 oxidation, and negatively correlated with MG-H1 modification, sLG content as well as GLO1 abundance of tumors (Fig. 5D, F). Higher expression of *GAPDH* correlates to worse ten-year survival across all cancer types, while the opposite is the case for *HAGH* (GLO2) and *LDHD* (PANCAN dataset; Figs. 4D and 5E). To additionally validate the connection between GAPDH and the GLO system, we both knocked down as well as pharmaceutically inhibited GAPDH in two different lung cancer cell lines and found that reducing active GAPDH in all cases enhanced both the expression as well as the activity of the GLO system, validating GAPDH as a master regulator of glucose metabolism and dicarbonyl stress, and with it, a critical drug target in lung cancer.

Lastly, in contrast to GLO1, we found GLO2 protein levels to be downregulated and its precursor sLG to be upregulated by hypoxia in lung cancer cells, suggesting that hypoxic cells need to detoxify MG either by alternative routes[47,70] or perhaps they rather accumulate the nontoxic intermediate sLG than producing higher levels of D-lactate, which would further increase acidification next to the already upregulated anaerobic L-lactate production from pyruvate.

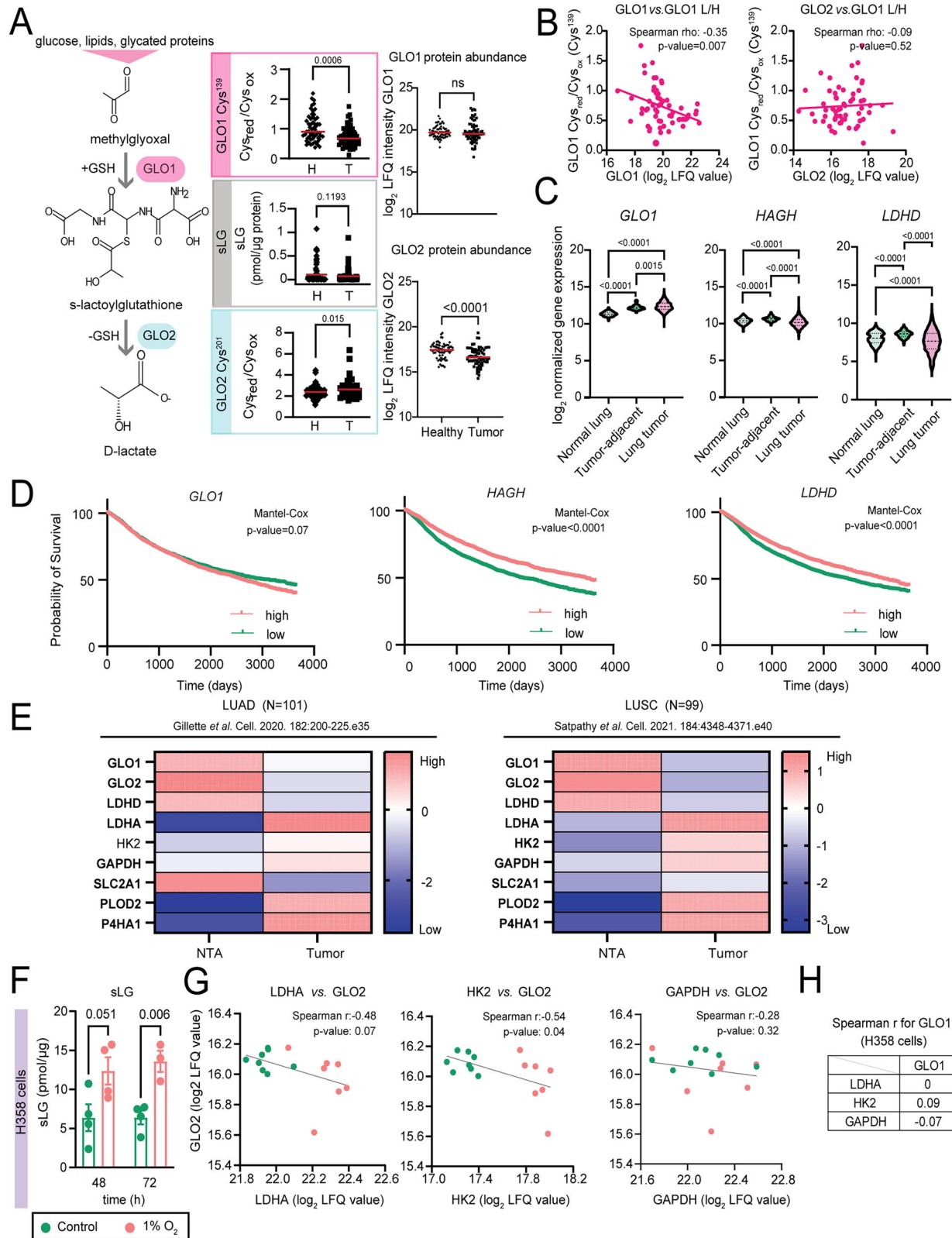

## Methods

### Study approval and ethical aspects

The use of human biomaterials was approved by the ethics committee of the Medical University of Graz (30-354 ex 17/18) and conformed with all pertaining regulations and the principles of the Declaration of Helsinki[71]. All participants provided written consent.

### Sample collection and patient information

Tissue pieces were collected in the span of one and a half years from curative surgery for NSCLC at the Division of Thoracic and Hyperbaric Surgery at the Medical University/State hospital of Graz, Austria. The cohort consisted of 64% male and 36% female patients with an average age of $65 \pm 9$ years at the point of surgery. The dominant form of NSCLC was lung adenocarcinoma (LUAD; 60%), followed by squamous

**Fig. 4 | Expression of GLO2 is lower in tumor tissue and correlates with worse survival outcome. A** Main methylglyoxal clearance pathway with redox status of GLO1's critical cysteine (Cys[139]; $N = 48$ patients, healthy or tumor tissue; two-sided paired Student's $t$ tests) as well as S-lactoylglutathione levels (sLG; the main GLO1 product; $N = 70$ two-sided, paired $t$ test; normalized on tissue protein content) in tumor versus healthy tissue. Contrary to GLO1, GLO2 (Cys[201]) was less oxidized in tumors ($N = 39$ patients, healthy or tumor tissue, two-sided paired $t$ test). On protein level, GLO1 was unchanged, while GLO2 protein was significantly less abundant in tumor ($N = 70$ (GLO1) and 66 (GLO2) two-sided paired Student's $t$ tests). **B** Correlation analysis of GLO1 oxidation (Cys[red]/Cys[ox] of Cys[139]) and GLO1 (left) or GLO2 abundance (right; in 59 (GLO1) or 58 (GLO2) tumors). **C** *GLO1* gene expression tends to be higher while *HAGH* (GLO2) and *LDHD* lower in lung tumor ($N = 1011$) versus normal lung ($N = 287$) and tumor-adjacent lung tissue ($N = 109$). All three

genes are upregulated in tumor-adjacent tissue compared to normal lung tissue (source: Xena browser; $N$ number of patients; two-sided $t$ test). **D** Kaplan–Meier plots of PANCAN dataset (10-year survival) for *GLO1*, *HAGH* (GLO2), *LDHD* ($N = 11,506$). **E** Target validation across two other pair-wise matched tumor vs. healthy proteomics datasets (lung adenocarcinoma (LUAD) ($N = 101$ per group; tumor or non-tumor adjacent (NTA))[16], and lung squamous cell carcinoma ($N = 99$ per group)[17] datasets). All significant targets upon two-sided pair-wise Student's $t$ test are bolded. **F** In H358 cells, hypoxia increased sLG ($N = 4$ biological replicates per condition and time point; two-sided, unpaired Student's $t$ test; mean ± SEM). **G** Rise in LDHA, HK2, GAPDH is inversely correlated to GLO2 (**G**) but not GLO1 (**H**) abundance in H358 cells ($N = 3–4$ biological replicates per condition and time point).

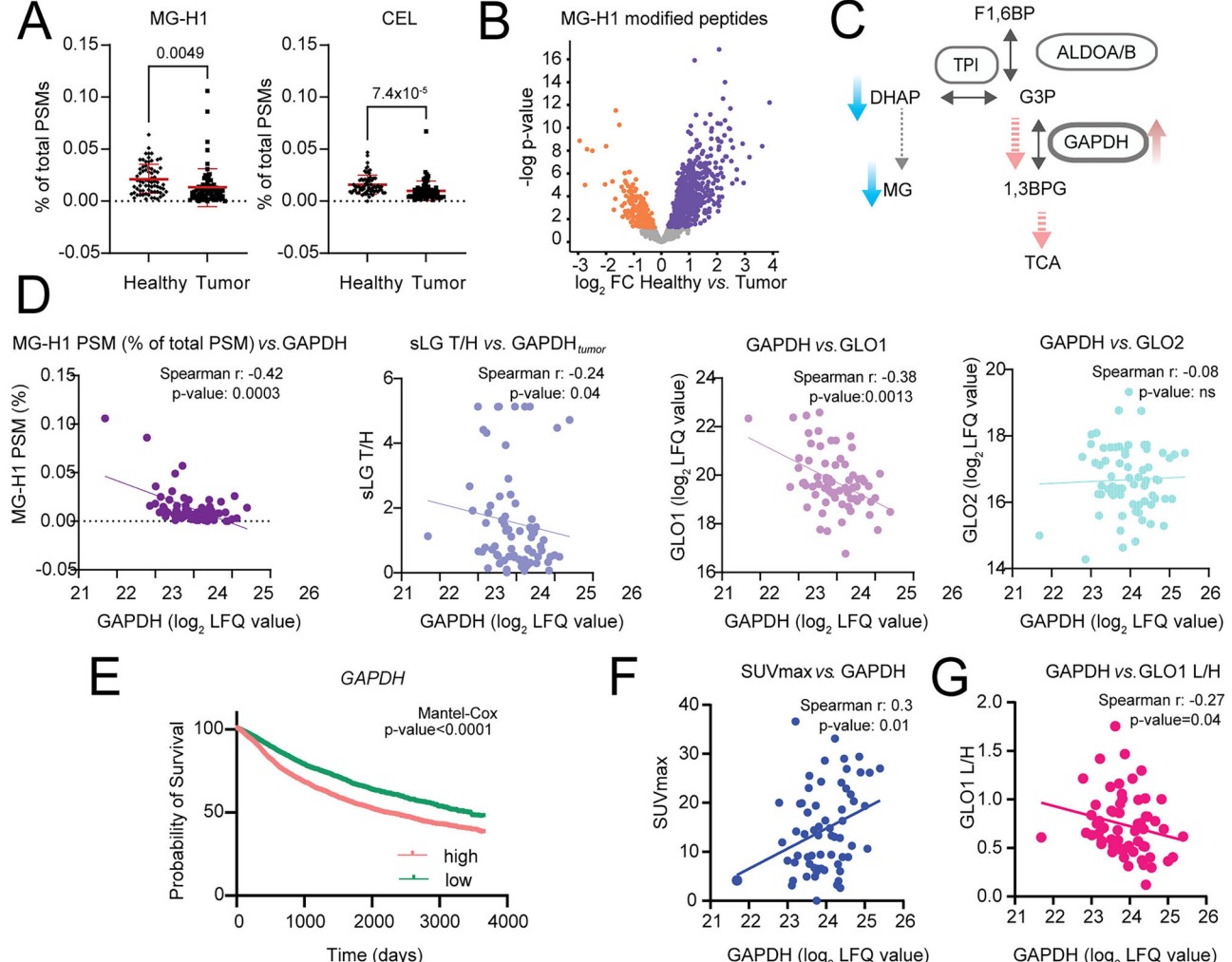

**Fig. 5 | GAPDH is inversely correlated with methylglyoxal production, GLO1 abundance and redox status. A** An open search of LFQ proteomics data detected a lower quantity of MG-driven protein modifications (MG-H1 on arginine and CEL on lysine) in tumor compared to healthy adjacent tissue ($N = 70$ patients, healthy or tumor tissue, two-sided paired Student's $t$ test). **B** A targeted search for MG-H1 modification corroborated the open search output ($N = 70$ patients, healthy or tumor tissue, two-sided paired Student's $t$ test). **C** GAPDH activity can control G3P and DHAP levels, which are precursors of MG. Correspondingly, **D** tumor MG-H1

modification frequency (as % of all PSMs), s-lactoylglutathione (sLG) values (as tumor/healthy (T/H) ratio) and tumor GLO1, but not GLO2 protein levels, inversely correlate with the abundance of GAPDH. Numbers of pairs per correlation analysis (i.e., number of patients): 69 (GLO2) and 70 (all other). **E** Kaplan–Meier plots of PANCAN dataset (10-year survival) for *GAPDH* ($N$ (number of patients) = 11,506). **F** GAPDH abundance was positively correlated with fluorodeoxyglucose (FDG) uptake of tumors and (**G**) with GLO1 oxidation levels ($N$ pairs for correlation analysis (i.e., number of patients): 63 or 59 correspondingly).

cell carcinoma (SCC; 31%). In 7% of the cases, no further pathological stratification at the point of sample collection beyond "NSCLC" or a mixed pathology was present (e.g., SCC and neuroendocrine tumor (NET)), and one case of large cell (LC) carcinoma was included. Individual patient information such as type of lung cancer, driver mutation

(when available), fluorodeoxyglucose (FDG) uptake (PET values, reported as SUVmax), gender and patient age, as well as smoker status, are listed in Supplementary Data 1. Molecular analysis covering mutation hotspots and gene fusions/gene skipping was carried out with either Ion AmpliSeq™ V2 Lung/Colon cancer panel (Thermo) or

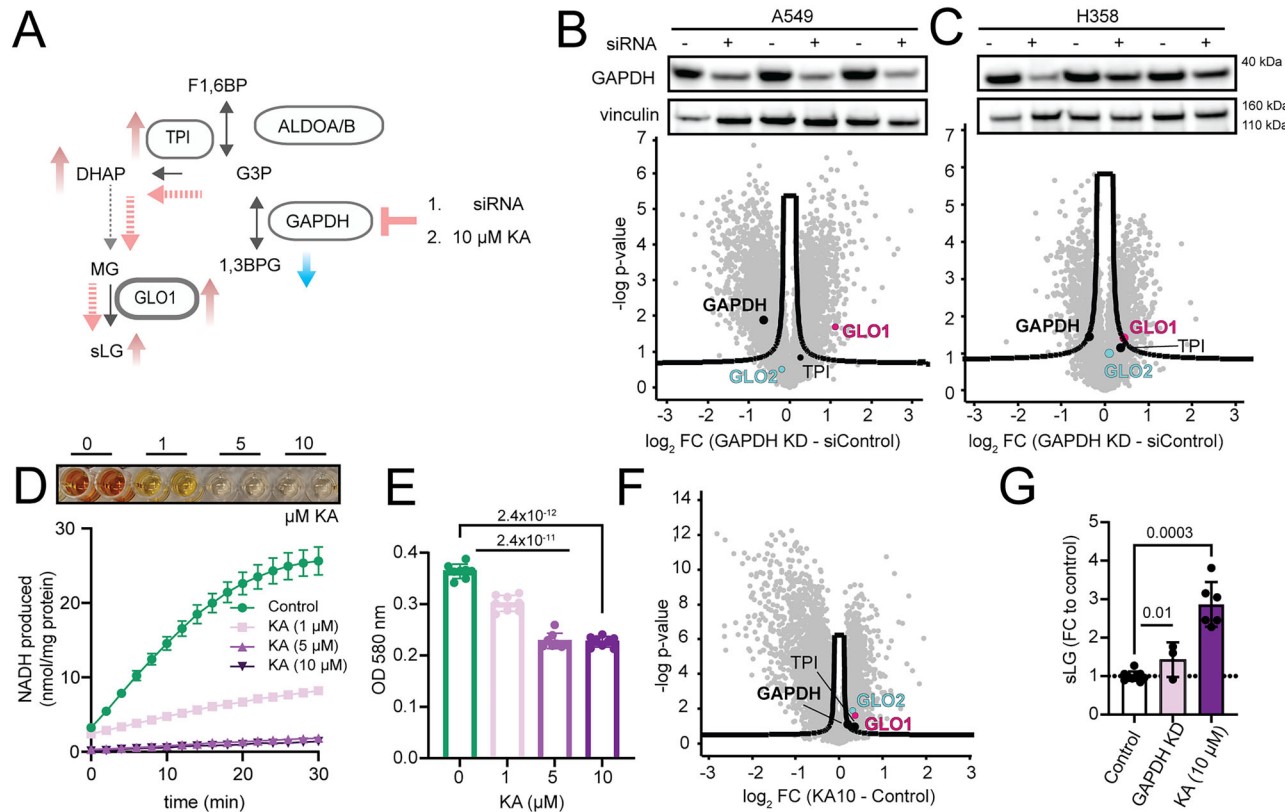

**Fig. 6 | Impaired GAPDH activity induces the glyoxylate system and increases sLG. A** GAPDH activity was reduced by siRNA-mediated knockdown or koningic acid treatment (KA; 10 μM). Inhibition of GAPDH is hypothesized to lead to G3P/DHAP accumulation, more MG production and induction of GLO. **B** (Top) Western blot confirmed siRNA-mediated GAPDH knockdown in A549 cells ($N = 3$ biological replicates per condition) and (bottom) consequent increase in GLO1 abundance displayed by volcano plot of quantitative proteomics data of GAPDH KD versus siRNA control ($N = 3$ biological replicates per group; unpaired multi-test corrected Student's $t$ test $p$ value < 0.05; S0 = 0.1). **C** Same as **B** only for H358 cells. For full scan blots, see the Source Data file. **D** GAPDH activity assay using 0, 1, 5, or 10 μM KA upon 24 h incubation with the drug confirmed abolished GAPDH activity by KA treatment ($N = 4$ biological replicates, each biological replicate represents a

mean of two technical replicates; graph: mean of biological replicates ± SEM). **E** Sulforhodamine B (SRB) proliferation assay of A549 cells after 24 h of 0, 1, 5, or 10 μM KA treatment ($N = 8$ biological replicates per condition; two-sided unpaired Student's $t$ test, mean ± standard deviation (SD)). **F** Volcano plot of proteomics analysis of A549 cells treated for 24 h with 10 μM KA ($N = 6$ biological replicates per condition, unpaired, two-sided multi-test corrected Student's $t$ test $p$ value < 0.05; S0 = 0.1). **G** sLG levels of GAPDH KD and KA-treated A549 cells represented as fold changes of their corresponding controls. For better overview, controls were merged into the same bar of the chart (GAPDH KD experiment: three biological replicates per condition; KA treatment: six biological replicates per condition; two-sided unpaired $t$ test; mean ± SD). In the figures, green represents control and purple rising KA concentrations or siGAPDH.

Archer Fusion Plex Lung Panel (ArcherDX) or by using targeted Idylla BRAF, KRAS and EGFR mutation kits (Biocartis). Of note, SCC cases were most often tested only for BRAF and or EGFR mutations. Tissue pieces were collected at the point of pathological examination (within the first hour upon surgery) and were immediately placed into sample tubes containing 1 ml of 80% methanol in 50 mM ammonium acetate and 2.5 mM N-ethylmaleimide (NEM; polar extract solution). Samples were homogenized with an Ultraturrax tissue homogenizer (IKA) and stored at −80 °C until further processing.

**Sample preparation for comprehensive redox analysis**
Homogenized tissue pieces were sonicated (2–5 s at 10% amplitude) in polar extract solution, then spun down to pellet proteins for 10 min at 16,000 × $g$. Supernatants were transferred to a new collection tube and dried down (for subsequent glutathione analysis) while the protein pellets were resuspended in 100 μl 50% trifluoroethanol (TFE)/50 mM ammonium bicarbonate (ABC) and processed for redox and quantitative proteomics.

**Proteomics sample preparation**
Tissue protein extracts in 50% TFE/ABC were subjected to protein estimation (BCA; Thermo), after which 50 μg of protein was aliquoted for further processing. In the next step, samples were diluted with

100 mM ABC, then reduced with 5 mM TCEP (in 50 mM ABC) for 30 min at 37 °C, re-alkylated with 10 mM d5-NEM, diluted with four volumes of 25 mM ABC and digested overnight with trypsin at 37 °C, (shaking at 550 rpm; 1 μg of trypsin per sample). In the last step, 4 μg of protein was desalted offline using in-house made stage tips (SDB-RPS, Empore-Supelco); and 300 ng of protein was injected into the LC-MS/MS system.

**Proteomics LC-MS analysis**
Chromatographic separation was performed using a Dionex Ultimate 3000 RSLC Nano system (Thermo Scientific) coupled with an Ionopticks Aurora Ultimate Series UHPLC C18 column (250 mm × 75 μm, 1.6 μm particle size). Mobile phase A consisted of 0.1% formic acid in water, while mobile phase B was acetonitrile containing 0.1% formic acid. The LC-MS/MS analysis ran for a total of 86.5 minutes following a gradient elution profile: 2% B from 0–5.5 min, 2–10% B between 5.5 and 25.5 min, 10–25% B from 25.5–45.5 min, 25–37% B over 45.5–55.5 min, and a ramp to 80% B from 55.5–85.5 min. This was followed by a hold at 80% B (65.5–75.5 min), a rapid return to 2% B over 75.5–76.5 min, and re-equilibration at 2% B until 86.5 min. The column was maintained at 40 °C with a constant flow rate of 400 nL/min. Mass spectrometric detection was carried out on a timsTOF Pro instrument (Bruker Daltonics) operating in positive ionization mode with trapped ion

mobility spectrometry (TIMS) active at full duty cycle. Both ramp and accumulation times were set to 100 ms. The electrospray source operated at a capillary voltage of 1600 V, with the dry gas set to 3 L/min at 180 °C. Data were acquired in a data-dependent manner using the parallel accumulation–serial fragmentation (PASEF) method. Precursor ions within the 100–1700 m/z range were selected, and each cycle included four PASEF ramps, yielding a total cycle duration of 0.53 seconds.

### Proteomics data processing and statistical analysis

All proteomics analysis included the same 70 samples per group (healthy and tumor). Raw data files were processed as previously described[72] with some modifications. Data analysis including database search and label-free protein quantitation (LFQ) was carried out with MaxQuant (v2.1.3.0)[73,74]. Here, methionine oxidation, NEM and d5-NEM on cysteine were selected as dynamic modifications and no static modifications were defined. Trypsin was set as a digestion enzyme, allowing for two missed cleavages. Redox proteomics was processed with FragPipe (version 18.0 containing MSFragger v.3.5[75] and IonQuant 1.8.0[76]). Open search as well as targeted search for methylglyoxal modifications were also carried out with FragPipe (same version or 19.0 (containing MSFragger 3.6 and IonQuant 1.8.9)). All data was searched against a Uniprot human protein database fasta file with 20398 entries downloaded on 15.08.2022, except for the MG-H1 search for which a fasta file containing 20806 entries (downloaded on 16.03.2023) was used. All outputs were filtered at 1% FDR for identification. If not stated otherwise, Perseus (v1.6.14.0) was used for downstream statistical analysis of resulting quantitative data[77]. All raw data, including search engine output have been deposited to PRIDE[78] with the following identification: PXD052340.

**Label-free (quantitative) proteomics.** For LFQ analysis, at least two peptides per protein were required for quantification. The match between run features was enabled with a matching time window of 1 min and an alignment time window of 20 min. In Perseus, the table with protein LFQ intensities was filtered for non-contaminants, then for at least 60% of valid values in at least one of the groups (healthy or tumor). Values were additionally normalized by subtracting the median value per column, and the missing values were imputed from a normal distribution (width 0.3, downshift 1.8), before carrying out multi-test corrected Student's t-testing between the groups (healthy *versus* tumor; FDR 5%, S0 = 0.1). The results of the LFQ proteomics of the patient data are reported in Supplementary Data 2. Significantly more or less abundant proteins were further subjected to GOBP using StringApp (v1.6.0)[79] and were additionally visualized using EnrichmentMap (v3.3.1)[80], all in Cytoscape v3.9.1[81]. For an easier overview of the data, EnrichmentMap output was additionally manually curated by grouping terms based on the type of biological process (see Suppl. Fig. S1). All individual terms encompassing such a "cluster" of GOBPs are reported in Supplementary Data 5 (with proteins significantly less abundant in cancer as input for enrichment) and Supplementary Data 6 (proteins significantly up in cancer as input for enrichment).

**Redox proteomics.** For peptide L/H (light to heavy, NEM to d5-NEM) ratio estimation, NEM and d5-NEM were configured as a "light" (NEM) and "heavy" (d5-NEM) label pair in IonQuant. Peptide lists with calculated ratios were then processed through an in-house Python script that first annotated the cysteine residues and then calculated a median value of peptides containing the same cysteine residue. The script was deposited on GitHub and made publicly available (https://github.com/BernhardRetzl/cysteine_label). Statistical analysis of obtained light/heavy (L/H) ratios for each cysteine (resembling ratios of reduced to oxidized cysteine residues) was performed in Perseus. There, the matrix was further filtered to contain only those cysteines with reported L/H ratios in at least 30% of the samples of each of the groups

(healthy or tumor). On the resulting matrices, a two-tailed Student's t test was performed between healthy and cancer groups (FDR corrected p value < 0.05 or non-FDR corrected p value < 0.05). The output of the redox proteomics analysis is reported in Supplementary Data 3, which includes the filtered output of the script (median value of the reported ratios per cysteine after filtering for valid values) as well as raw output from FragPipe containing all the detected cysteine-containing peptides. Proteins whose oxidative state was significantly altered (p value < 0.05) were further used for subcellular annotation using Uniprot ID mapping as well as GO enrichment analysis of GOBP using the STRING database web-platform (v12.0). The cut-off for all enrichment analysis was always FDR corrected p value < 0.05. Subcellular localization information, as well as GOBP enriched terms from the redox dataset are reported in Supplementary Data 4.

**Open search.** We conducted an open search utilizing FragPipe's default settings[82], adjusting the precursor mass tolerance to a range of −150 to +500 Da. Dynamic modifications included methionine oxidation and both NEM and d5-NEM modifications on cysteine residues. The digestion enzyme was specified as strict trypsin, allowing up to two missed cleavages. PTM-Shepherd was employed to identify post-translational modifications by referencing the Unimod database and normalizing the results based on peptide spectral matches (PSMs). A minimum of 10 PSM was set as a threshold to be required for the detection of a modification. The modified precursor tolerance was set to 0.01 Da, and the peak picking width to 0.002 Da. Results of the open search are listed in the Supplementary Data 7.

**MG-H1 directed search.** A methylglyoxal hydroimidazolone 1 (MG-H1) directed search was carried out according to the default parameters for a closed search in MSFragger with the following adaptations: precursor match tolerance was set to ± 20 ppm, strict trypsin was used as enzyme but since MG-H1 is expected to be on arginine affecting tryptic cleavage efficiency, missed cleavage allowance was increased from two to three. Here, methionine oxidation, NEM and d5-NEM on cysteines and MG-H1 (+54.0106 Da) on arginine and lysine were selected as dynamic modifications, and no static modifications were defined. The obtained table of quantified peptides was then imported into Perseus (v2.0.1.0) and filtered to contain only the peptides with reported MG-H1 modification on arginines. The resulting matrix was further filtered to contain only those MG-H1 peptides with reported values in at least 10 samples in at least one group (healthy or tumor) on which then the GO annotation and statistical analysis was carried out (Student's t test multi-testing corrected p value < 0.05, S0 = 0.1). The results of the MG-H1 directed search are also reported in Supplementary Data 7.

### Glutathione and small molecular thiols analysis

Glutathione ratio analysis was carried out by our previously described method[1,2] with some modifications. Briefly, dried samples were resuspended in 100 μl of 50 mM ammonium acetate (AA), after which excess NEM was removed using 300 μl of dichloromethane. Consequently, 45 μl of the aqueous phase was transferred to a new sample tube prior to reduction with 2.5 μl of 50 mM TCEP in AA for 30 min at 37 °C and re-alkylation with 2.5 μl of 100 mM d5-NEM. Samples were diluted 1:4 with AA, of which 2 μl were subjected to liquid chromatography-tandem mass spectrometry (LC-MS/MS) using the LC-MS-8060 system (Shimadzu) with a flow rate of 0.2 ml/min and following gradient: 0-10 min: 1-16% B, 10.01-15.00 min: 1% B (A being 0.1% formic acid; B 0.1% formic acid in acetonitrile). MS was operated in positive mode, with gas flow rates of 2.5, 10 and 10 L/min for nebulizing, drying and heating gas, respectively. The desolvation line temperature was set to 250 °C and the heat block temperature to 400 °C. Other small molecular thiols were measured in a separate LC-MS run with exactly the same LC-MS parameters, only with

**Table 1 | Multiple reaction monitoring (MRM) transition table with parameters**

| | Parent m/z | Product m/z | Dwell time (ms) | Q1 pre bias (V) | CE | Q3 pre bias (V) |
|---|---|---|---|---|---|---|
| GSH | 308 | 179 | 65 | −11 | −19 | −23 |
| GSSG | 613 | 355 | 65.6 | −14 | −20 | −25 |
| GSH-NEM | 433 | 304 | 65.6 | −10 | −17 | −21 |
| | 433 | 201 | 65.6 | −16 | −21 | −21 |
| GSH-d5-NEM | 438 | 309 | 65.6 | −10 | −17 | −21 |
| | 438 | 206 | 65.6 | −16 | −21 | −21 |
| IS | 441 | 312 | 100 | −10 | −17 | −21 |
| | 441 | 206 | 100 | −16 | −21 | −21 |
| sLG | 380 | 223.05 | 100 | −11 | −18 | −24 |
| | 380 | 148.1 | 100 | −11 | −22 | −30 |
| | 380 | 76.05 | 100 | −11 | −37 | −14 |
| CysGly-NEM | 304 | 287 | 65.6 | −11 | −14 | −19 |
| | 304 | 201 | 65.6 | −11 | −15 | −21 |
| CysGly-d5-NEM | 309 | 292 | 65.6 | −11 | −14 | −19 |
| | 309 | 206 | 65.6 | −11 | −15 | −21 |
| Cys-NEM | 247.11 | 230.05 | 65.6 | −17 | −14 | −24 |
| | 247.11 | 201.05 | 65.6 | −14 | −13 | −21 |
| Cys-d5-NEM | 252.11 | 235.05 | 65.6 | −17 | −14 | −24 |
| | 252.11 | 206.05 | 65.6 | −14 | −13 | −21 |
| GluCys-NEM | 376 | 247 | 65.6 | −14 | −15 | −35 |
| | 376 | 201.1 | 65.6 | −14 | −20 | −20 |
| GluCys-d5-NEM | 381 | 252.05 | 65.6 | −14 | −15 | −25 |
| | 381 | 206.1 | 65.6 | −14 | −20 | −20 |
| HCys-NEM | 261 | 56.05 | 65.6 | −13 | −16 | −22 |
| | 261 | 215.05 | 65.6 | −13 | −12 | −23 |
| HCys-d5-NEM | 266 | 56.05 | 65.6 | −13 | −16 | −22 |
| | 266 | 222.05 | 65.6 | −13 | −12 | −23 |

*CE* collision energy, Q-quadrupole (*GSH* glutathione, *GSSG* glutathione disulfide, *NEM* N-ethylmaleimide, *IS* internal standard ($^{13}C^{14}$,N-GSH-d5-NEM), *sLG* S-lactoylglutathione, *CysGly* cysteinyl glycine, *GluCys* gamma glutamyl-cysteine, *Cys* cysteine, *HCys* homocysteine).

different MRM transitions. A list of all transitions and their corresponding voltages, energies and dwell times is presented in Table 1. Peak integration was carried out in Shimadzu Post-run Analysis. All analytes were normalized on both internal standard ($^{13}C^{15}$,N-GSH-d5-NEM) as well as protein content of the corresponding sample.

### Cell experiments
Human lung cancer cell lines A549 and H358 were obtained from either Cell Lines Service (Eppelheim, Germany; A549 cells, 300114) or the American Type Culture Collection (ATCC, Manassas, VA, USA; H358 cells; CRL-5807) and cultured in RPMI 1640 (R0883; Sigma) supplemented with 10% fetal bovine serum (FBS; Gibco), 5 mM glutamine and 5 mM Pen Strep, and sub-cultured regularly.

**Hypoxia incubation.** 300,000 H358 cells were seeded in sixwell plates in quadruplicates per plate and placed either at 1% or 21% oxygen for either 48 or 72 h, respectively. Upon the indicated time, cells were

harvested in 500 μl of harvesting solution (80% methanol in 50 mM AA, supplemented with 2.5 mM NEM and heavy glutathione (GSH) internal standard), sonicated for 5 s at 10% amplitude and processed by our published one-pot redox metabolite and protein thiol analysis approach[15]. Briefly, samples were spun down for 15 min at 15,000 × g at 4 °C, after which they were separated into two phases: the upper for thiol analysis and the pellet for proteomics. The upper phase was processed as described in the Glutathione analysis and the pellet as described in the Proteomics sample preparation subchapters. Each time point included its corresponding 21% oxygen control. Raw data from proteomics analysis as well as output of the processing was deposited to PRIDE with the following identifier: PXD062503.

**GAPDH inhibition and knockdown proteomics.** For siRNA-mediated knockdown of GAPDH, 150,000 A549 and H358 cells were seeded in triplicates of a six-well plate and transfected upon reaching ca. 80% confluency with a silencer™ siRNA kit targeting GAPDH or a non-target control (AM4605; Thermo), using RNAiMax (Thermo) and according to the manufacturer's instructions for a sixwell plate scale. For GAPDH inhibition, 150,000 A549 and H358 cells were seeded in six replicates per condition, then treated with 10 μM Koningic acid (KA) or vehicle control (DMSO). 48 h post-transfection or 24 h post KA treatment, cells were harvested in 500 μl of the aforementioned harvesting solution and processed according to our one-pot redox metabolite and protein thiol analysis approach.

**Immunoblotting GAPDH.** Methanol precipitated protein pellets were dissolved in 50% TFE/ABC; from which, upon protein estimation, 20 μg was mixed with 1× loading buffer, processed for SDS-PAGE and consequently transferred to PVDB membrane using an iBlot system (Thermo). Membranes were blocked for 1 h in protein-free blocking buffer (Thermo), then incubated overnight with 1:1000 diluted anti-GAPDH mouse monoclonal antibody (60004-1-IG; Proteintech) or anti-vinculin mouse monoclonal antibody (sc-73614; Santa Cruz). The following day, membranes were washed and consequently incubated with anti-mouse horseradish peroxidase (HRP)-conjugated secondary antibody in 1:10,000 dilution (G21040; Invitrogen). The resulting blots were visualized with SuperSignal West Pico enhanced chemiluminescence reagent (Thermo) on Chemidoc (Biorad).

**GAPDH activity assay.** For GAPDH activity assay, 300,000 A549 cells were separately seeded per well of a 6-well plate in quadruplicates per condition, with conditions being: 1 μM, 5 μM, 10 μM KA treatment or DMSO control. Upon 24 h incubation with KA, cells were harvested and subjected to analysis using a glyceraldehyde-3-phosphate dehydrogenase activity assay kit (ab204732; Abcam) according to manufacturer's instruction. The assay was additionally carried out in technical duplicates for each biological quadruplicate.

**Sulforhodamine B assay.** Cytotoxic effects of KA were monitored using Sulforhodamine B colorimetric assay as previously described[83]. Briefly, 5000 A549 cells were seeded in eight replicates per condition, then treated with either 1, 5, 10 μM KA or vehicle control (DMSO). 24 h after the treatment, cells were fixed with 10% trichloroacetic acid, extensively washed, then incubated with the dye, washed and finally resolubilized in 10 mM Tris base, prior to scanning the plates at 565 and 690 nm (main signal and background, respectively).

**Proteomics of GAPDH knockdown and inhibition.** For proteomics of GAPDH KD and KA-treated cells, protein pellets upon methanol precipitation were also resuspended in 50% TFE/ABC and 20 μg protein was further subjected to redox proteomics preparation as previously described. LC-MS/MS was carried out on 500 ng of injected material similar as to aforementioned Proteomics LC-MS analysis with following adaptations: the timsTOF HT mass spectrometer (Bruker Daltonics)

was operated in positive mode with enabled trapped Ion Mobility Spectrometry (TIMS) at 100% duty cycle (100 ms ramp and accumulation time). Scan mode was set to data-independent parallel accumulation–serial fragmentation (diaPASEF) using parameters previously described[84]. In brief, 21 isolation windows of 25 m/z width spanning from m/z 475 to 1000 were defined. After an MS1 scan, 2–3 isolation windows were fragmented per TIMS ramp from both sides of the mass range (e.g., m/z 400–425 and 800–825). The collision energy was set to rise linearly starting from 20 eV over the covered mobility range (for 1/K0 values between 0.85 and 1.27). The total resulting DIA cycle time was estimated to be 0.95 s. The resulting raw data were processed in DIA-NN[85] (v1.9.2) against a fasta file containing 20806 entries (including contaminants). Methionine oxidation was set as variable modification and no fixed modifications were set. Trypsin was used as a digestion enzyme with one missed cleavage allowed. The match between run feature was enabled, and all of the outputs were filtered at 0.01 FDR. The resulting quantitative results matrix was imported in Perseus 1.6.14.0, filtered for contaminants, then for valid values (at least three or at least five valid values in each group of GAPDH KD or KA treatment experiments, respectively) and consequently processed for statistical analysis (FDR corrected $p$ value < 0.05, S0 = 0.1). H358 matrix was additionally normalized by median centering prior statistical analysis. All raw data, including search engine output has been deposited to PRIDE with the dataset identifier PXD061610.

## Statistics and reproducibility

The patient study included 70 participants. No statistical analysis was carried out to determine sample size - it was dictated by sample availability over the given time span. No samples were excluded. Cell experiments were carried out either on three or four biological replicates per condition and time point (hypoxia; 21% or 1% oxygen and 48 or 72 h, respectively); or three biological replicates per condition (siRNA control or siGAPDH; two different cell lines) or six replicates per condition (GAPDH inhibition; DMSO control or 10 μM koningic acid; two different cell lines) or eight replicates (SRB assay). GAPDH western blot was repeated twice and corroborated by proteomics experiments. In case of correlation of hypoxic markers with GLO1/2 abundance in H358 cells (Fig. 4G, H) one sample from 1% oxygen/48 h condition was excluded due to overall low protein quantitation values. The data is nevertheless reported in the Supplementary Data 8.

Data in Supplementary Data 2 and 8 were subjected to two-sided, unpaired FDR corrected Student's $t$ tests. Data in Supplementary Data 3 was subjected to both multi-test corrected as well as not corrected two-sided, unpaired Student $t$ tests. Enrichment in Supplementary Data 5 and 6 was subjected to FDR corrected enrichment analysis using StringApp in Cytoscape as described. Everywhere else, for patient data, two-sided paired $t$ tests were carried out, while for in vitro data, it was subjected to unpaired two-sided Student $t$ tests, always with a $p$ value of 0.05 as the cutoff for significance.

## Reporting summary

Further information on research design is available in the Nature Portfolio Reporting Summary linked to this article.

## Data availability

All data generated in this study are provided in the Article, Supplementary Information, Supplementary Data and Source Data file. All raw proteomics data were submitted to PRIDE repository with following accession numbers: PXD052340, PXD062503 and PXD061610. Raw exports of targeted metabolite measurements are included in the Source Data File. Source data are provided with this paper.

## Code availability

The code used for cysteine residue annotation is publicly available on GitHub.

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

## Acknowledgements

This research was funded in whole or in part by the Austrian Science Fund (FWF) [10.55776/F73, 10.55776/W1226, 10.55776/FG12, 10.55776/COE7, 10.55776/COE17] to R. Birner-Gruenberger, the City of Vienna [H-867676/2022] to T. Tomin and the Marietta Blau fellowship to S. E. Honeder. Additionally, our thanks go to Shimadzu Austria for supporting this research through instrument access given in the Metabolomics and Bioprocess Analytics laboratory (TU Wien). For open access purposes, the author has applied a CC BY public copyright license to any author accepted manuscript version arising from this submission.

## Author contributions

T.T., M.S., and R.B.G. devised the study and wrote the manuscript. T.T. and D.G. carried out the experiments. L.B. and J.L. provided the samples and carried out pathological analyses. S.E.H. and L.L. collected and prepared the samples for redox analysis. T.T. and S.E.H. did the data analysis and statistical evaluation. B.R. contributed to the data and statistical analysis.

## Competing interests

The authors declare no competing interests.
