## [Transparent Peer Review file · Nature Communications]

Increased antioxidative defense and reduced advanced glycation end product formation by metabolic adaptation in Non-Small-Cell-Lung-Cancer patients

Corresponding Author: Professor Ruth Birner-Gruenberger

Version 0:

Reviewer comments:

Reviewer #1

(Remarks to the Author)

Tomin et al. submitted a manuscript describing a specific cross-talk between ROS and metabolism in the context of Non-small cell lung cancer. For this they employ for the first time an original approach by comparing redox proteomics of tumour tissue and matched healthy tissue. They found changing levels of protein and oxidation of several enzymes in pathways related to glycolysis, mitochondrial metabolism, and reactive oxygen species. Among all the expression and oxidation changes in glucose metabolism, TCA cycle etc, the relative modifications on the metabolism of methylglyoxal are of particular interest.

While some of the findings are expected, like the increase oxidative stress in tumours compared to healthy tissue, the (presumably) hypoxia-related reduction of toxic byproducts of glucose metabolism (like MG or AGEs) can help tumour cells to grow in difficult conditions. Previous reports (Luengo et al, ref 57 in the manuscript) analysed the contribution of methylglyoxal pathway enzyme GLO1 in NSCLC and its possible therapeutic use. The manuscript is mostly a description of novel metabolic traits of NSCLC with potential interest for this and other cancers although the scope of the main findings is limited by the complex validation of these results in other patient cohorts of disease models. In addition, some relevant points raised up by the results were not confirmed and remain rather speculative.

Some major concerns need to be addressed before the article can be considered for publication.

Major:

- Information on the clinical characteristics of the patients and of the samples is rather scarce in methods and results and could be relevant for the analyses and conclusion of the study. Although some disease biomarkers might not be regularly analysed in the clinical management of early-stage NSCLC, some other could be easily analysed on the resected sample.

- Resectable NSCLC is a clinical entity that includes different early NSCLC stages. Information of the tumour stage should be included in supp.

- As mentioned in the text, some genetic alterations impact metabolism (i.e. KRAS). Are driver mutations known?

- Stromal cells (immune and others) may rely on the same metabolic traits and oxidative stress in an inflammatory context like a tumour. Is there any data on the sample cellularity? Otherwise, tumour cell (or even immune cell) density should be measured to identify any possible bias on the bulk proteomic analyses.

- Absence of similar approaches prevents validation of the oxidation data on independent cohorts. However, protein levels of enzymes could be in other NSCLC patient cohorts subjected to proteomic analyses. Lethiö et al (Nat Cancer 2021) published MS-based proteomic data, together with lots of other clinically relevant annotations, coming from NSCLC samples (mostly resectable early-stage). Since all the data comes from a single series of samples, relevance, impact and robustness would significantly improve if some results can be further validated beyond the associations between GLO2 and HIF1a (very valuable and necessary tough).

- This association between hypoxia and reduced GLO2 is mentioned in the conclusions but remains speculative. This conclusion would require to be solidify with some more experimental data (even in vitro) by inducing hypoxia and studying the expression of these enzymes. Additional data that might support this could be produced by analysing HIF1a direct targets like LDHA or HK2 (increased in tumours compared to healthy tissue). These can be discussed as surrogate markers

of HIF1a activation in your same experimental samples. Same happens with the MG, which is suggested to be either less formed or more secreted. This could be experimentally addressed to be part of the conclusions.

Minor:

- Abstract could make more emphasis on the relevance of the study in the context of cancer/NSCLC and the problematic trying to get solved.
- For a better understanding of non-experts in oxidation and metabolism, how representative is thiol oxidation of overall oxidation?

(Remarks on code availability)

Reviewer #2

(Remarks to the Author)

This manuscript is a proteomic analysis of 70 lung tumor samples and 70 matched healthy lung samples, looking at protein levels as well as levels of cysteine oxidation. In addition, the authors look at protein glycation caused by the reactive metabolite methylglyoxal. Overall, this is a very nice dataset which will certainly be useful for the community. In particular, the identification of many oxidized cysteine residues in glycolytic enzymes is exciting and could serve as a starting point for future, mechanistic studies. The current work, however, is entirely descriptive, so it could fit well as a resource paper.

As described below, I believe some of the biological conclusions / interpretations may not be warranted by the data, and would need to be changed:

1. Fig 3D- If I understand correctly, this is a correlation analysis between HIF1a mRNA levels and Glo1/2 mRNA levels, and the authors conclude this is “suggesting a strong, hypoxia driven in vivo down-regulation of GLO2 gene expression”. However, hypoxia influences HIF1a protein stability, not mRNA as far as I know? In which case this conclusion may not be warranted. I’m not sure what HIF1a mRNA levels say about HIF1a activity, since it’s mainly post-translationally regulated?

2. Fig 3F – if tumors have lower levels of MG-H1 and CEL, why do the authors conclude in the corresponding Results section that this “might indicate a higher presence of MG in the blood stream of lung cancer patients” ? Wouldn’t it be the other way around? Alternatively, if the authors are interpreting lower MG levels in the tumor to indicate that more MG is being excreted by the tumor, this would require knowing that the rate of MG production is similar in control tissue and tumor tissue? (in particular since I don’t know of any mechanisms of active MG excretion, in which case MG excretion might not be a process that can be regulated?)

3. Line 468 “In line with ... a potential increase in MG excretion by the tumor, we observe a stark upregulation of RAGE” Besides the comment #2 above, isn’t RAGE activation also post-translational? Would elevated levels of AGEs lead to elevated levels of RAGE protein? If not, it’s not clear that elevated levels of RAGE protein indicate elevated AGE levels. (Also, where does one see elevated RAGE protein levels in the figure?)

4. One of the main conclusions in the abstract is that “cancer strives to maintain oxidative metabolism amid the rise of intracellular oxidative stress”.
What is the evidence for elevated oxidative stress in the tumor samples, if the tumors samples and healthy tissue samples did not separate from each other on a PCA based on the cysteine oxidation (Suppl. Fig. 1C) and the Cysred/Cysox ratio of all samples was clustered around 1 for both healthy and tumor samples? Of the cysteine residues that showed differential oxidation (Fig. 1A), there’s roughly an equal number that go up or down in the tumor samples compared to healthy tissue. Wouldn’t this suggest specific regulation rather than a global change in oxidative stress?
One could argue that if tumor samples have equal levels of global cysteine oxidation compared to healthy tissue, but elevated levels of antioxidant proteins, then they have a combination of elevated oxidative stress and elevated antioxidant activity, resulting in equal levels of cysteine oxidation. But instead, tumor tissue has lower levels of antioxidative proteins compared to healthy tissue (Fig. S1E). So this would rather speak against elevated oxidative stress in the tumor samples.

So overall, this central claim does not seem to me to be supported by the data.

Minor Issues:

1. Line 229 “while almost three times more significantly more oxidized proteins”
Extra “more”

2. Fig 1E – what does “Genes (%)” mean? I assume that’s the percent of genes in that category that are in the gene set?
Please specify.
The figure should also show FDR or corrected p-values on the bars.

3. Line 379 “we addressed levels of sLG,”
We assessed levels ?

(Remarks on code availability)

Reviewer #3

(Remarks to the Author)

Oxidative modification of cysteines in proteins has emerged as a significant cell regulatory mechanism. Cancer cells use multiple mechanisms to survive through oxidative stress, in part through oxidation-mediated modification of protein activity. Therefore, identification of oxidised proteins is an important step for understating mechanisms that drive cancer cell survival. Several studies have reported methods to identify oxidatively modified proteins in cultured cells, but it has been unclear whether the same proteins, and to which extent, are oxidatively modified in tumour tissues. In this manuscript, Tomin et al. address this point by using a method they previously developed to identify proteins that carry oxidatively modified cysteines in normal and tumour tissue from human non-small cell lung cancer (NSCLC) patients.

The authors show that approx. 10%-30% of the quantified oxidised cysteines are differentially oxidised in tumour vs. paired non-tumour tissue, and several of the proteins carrying these cysteines can be linked to major oncogenic pathways. Furthermore, the authors provide evidence that oxidation in several proteins involved in central carbon metabolism is higher in tumours compared to non-tumour tissue. Tomin et al. also provide evidence that proteins of the glyoxylate pathway are significantly enriched among those that are more oxidised between tumour and non-tumour tissues. The glyoxylate pathway breaks down methylglyoxal, a toxic metabolic byproduct of central carbon metabolism. Pivoting on this observation, the authors provide evidence that a combination of cysteine oxidation and changes in the abundance of glyoxylate pathway enzymes may lead to increased methylglyoxal levels in tumours. However, methylglyoxal adducts are higher in surrounding tissue compared to tumours, suggesting that tumours deal with glyoxylate system inactivation by increased excretion of methylglyoxal.

Overall, while evidence of causal links between the various observations the authors describe is missing, the main strength of this study is the novelty of analysing protein oxidation status in human tumours. The following points may help strengthen the author's claims.

1. Lines 385-405 and discussion lines 506-524: The authors' observation that low GLO2 levels correlate with poorer patient survival supports the idea that loss of glyoxylase system activity promotes tumour growth, however it also raises the possibility that suppressed GLO2 protein expression may suffice to deactivate this pathway. Given that the backbone of this study is the identification of oxidised proteins in patient samples, further investigation of the context in which GLO1 oxidation may be relevant is needed. To address this issue, the authors could consider further analyses of their dataset:

- test whether GLO1 oxidation is more likely to occur in tumours with high GLO2 levels (thereby showing that GLO1 oxidation is an alternative route for inactivating the pathway when GLO2 is present).
- Furthermore, to strengthen the proposed link between the glyoxylase enzyme oxidation status and hypoxia, can the authors assess how well hypoxic markers in the 70 tumours they analysed correlate with GLO1 oxidation?

2. Line 406-onwards and related to the last point in #1: The authors use HIF1 α mRNA expression as an indicator of hypoxia, however, the main regulatory mechanism of hypoxia response is post-transcriptional stabilisation of HIF1 α . HIF1 α gene targets (such as CA9) would be more suitable for inferring hypoxia and the authors should test the correlation between HIF1 α targets and GLO1/2.

3. In Fig. 1B, the authors calculate enrichments of the identified oxidised proteins according to their cellular location; they then use these enrichments to argue that the location of oxidised proteins differs between tumour (T) and non-tumour (NT) tissues. However, changes in the T vs NT proteome are likely to affect (and, indeed, may account for) these enrichments. Apologies if I missed this, but I did not find information indicating that the authors took into account differences in the background proteomes of T and NT for these enrichment calculations, which is essential in order to provide meaningful enrichment statistics.

4. Line 299: the authors attribute lack of GSH/GSSG ratio differences to increased GSH synthesis. This idea would be better supported by calculating the sum of GSH and GSSG abundances (as an indicator of the total glutathione pool size), which, given the increased GSSG, is likely to be found increased, in support of the authors' claim. Presenting these data is important, as, increased GSH synthesis enzyme gene expression (shown in Fig. S3B) correlates with but does not prove increased GSH synthesis. Furthermore, given that the NRF2 pathway is important in tumour antioxidant responses and may drive glutathione synthesis, the authors may want to investigate whether their proteomic or redox data (the NRF2 regulator Keap1 carries regulatory reactive cysteines) for evidence of NRF2 pathway activation.

5. Furthermore, while glycolytic intermediate re-routing, as the authors propose, may contribute to the increased glutathione reduction to produce GSSG, increased glutathione synthesis per se could also be explained by divergence of glucose-derived carbons into serine/glycine synthesis. The authors should discuss this possibility.

Other points:

- Line 462: define "AGEs" acronym.

- Fig. 3A: sLG graph y axis "pmol/mg": does the mass unit refer to tissue protein or dry/wet tissue weight?

(Remarks on code availability)

Reviewer #4

(Remarks to the Author)

In the manuscript entitled "Active Oxidative Metabolism and Impaired Glyoxalase System Under Increased Intracellular Oxidative Stress in Non-Small Cell Lung Cancer" Tomin and colleagues performed comprehensive evaluation of proteome and redox proteome of non-small cell lung cancer in comparison to control (adjacent tissue). The authors purpose the impact of protein redox modulation in metabolic rewiring of tumor cells, especially of glyoxalase system. Complementary analyses such as estimation of glutathione content and database search were done. In conclusion authors report that both redox and protein abundance changes contribute to the rise of intracellular oxidative stress and deactivate glyoxalase system which is compensated by higher excretion or lower production of methylglyoxal.

In summary, the manuscript is well written, contributes important results to the field of redox biology and provides novel insights into the role of redox metabolism in lung cancer. However, the manuscript contains several uncertainties, the clarification of which would contribute significantly to the strength of the claims made and to the reproduction of the methods.

Major comments

1. The healthy tissues were dissected from the different part of the lung of the same patient. Authors should demonstrate the definition/picture/cytology of non-cancerous changes in the healthy tissue. Are the ratios of protein abundances between healthy and tumor tissues done within the same patient?
2. It is surprising that the ratio between oxidized and reduced peptides is 1 in both tissues. According to the previous publications the median peptide oxidation is ranging between 5-20%. Can authors explain such discrepancy?
3. In Fig. 1B authors showed that tumor tissues in contrast to healthy tissue "have relatively more oxidation-affected intracellular proteins". The analysis involved both significantly reduced and oxidized peptides. It would be more correct to use the term "redox changed" peptides. Separate GO analysis for the reduced and oxidized peptides would clarify the demonstration.
4. In line 284 authors claim that they showed for the first-time large number of glucose-related metabolic enzymes are susceptible to redox modulation in vivo, please see the papers of Xiao et al. (PMC8164166) or Pimkova et al. (PMC9157258).
5. In Figure 2, some of the enzymes such as HK2 is demonstrated as the one with unaffected redox state, but searching in Table S3 it was not identified at all. It would be more accurate to distinguish between these findings.
6. Line 379, to be able to claim if the activity of GLO1 is lower both the substrate (MG) and the product should be determined.
7. Were GLO1 or GLO2 shown as targets of HIF1A according to available databases (e.g. <https://guolab.wchscu.cn/hTFtarget/#1/>)
8. In TCGA analysis, the GLO1 and GLO2 expression is evaluated in three types of lung tissues: healthy, tumor-adjacent, tumor. Which of them are relevant to the analyses done by authors? For the simplifications only the TCGA analysis of compared tissues would be good to show.
9. It would be useful to show the summary of open search analysis results, such as how many proteins were identified and how many of those were modified with MG.
10. In Fig. 3F it seems that majority of identified peptides is MG-H1 modified. Are indeed such many peptides MG-H1 modified?
11. Authors point to the increased glycolytic activity of lung tumor tissue; may they demonstrate it?
12. Authors conclude that both the redox change of GLO1 or protein abundance of GLO2 contribute to the phenotype of tumor cells. It would be very helpful if e.g. using lung cell line authors could conclude which of those two conditions contribute to the phenotype. For example, by modulation of redox environment using antioxidants or overexpression of GLO2.
13. Are available data of known mutations present in analyzed tissues? This is important to exclude the possibility of enzymes deactivation by mutation.

Minor comments

1. Inconsistency in labeling of Supplementary figures.
2. There are no links to Supplementary figure 1A,1B 1E.
3. In Fig. 1A, it is more common to present ratio of oxidized vs. reduced and tumor vs. healthy. The Fig. would be easier to read when authors demonstrate e.g. that RACK1 and CAV1 are the most oxidized proteins in tumors.
4. RACK1 and CAV1 proteins are the most significantly changed but not the most oxidized as written in line 207.
5. In Fig. 1A line 244 "each dot represents an individual cysteine residue". More correct would be individual peptide, since some of the peptides contain more than one cysteine.
6. The y axis marked as "-log FDR corrected p-value" according to supplementary table S3 rather shows the non-corrected p-values.
7. In Supplementary table 3, column "C" is probably not the logarithm as written in the header.
8. For the more fluent reading I would recommend to put the link to Fig. S2 and mention the GOBP analysis was done on

protein abundances and showed relevant biological processes to be enriched in line 276.

9. In line 402 it is unclear what "usual" means.

10. Line 462 shortage "AGE" not defined.

(Remarks on code availability)

Reviewer #5

(Remarks to the Author)

(Remarks on code availability)

Version 1:

Reviewer comments:

Reviewer #1

(Remarks to the Author)

Authors addressed successfully the raised questions and provided significant amount of new data that gives a better idea of a tumour metabolic rewiring to adapt glucose metabolism and oxidative stress in disease context. Now they have validated up to some extent the findings from patient samples, and open new avenues in this filed in NSCLC.

Here we provide some additional minor comments:

- Figure 4 needs revision, since text, caption and figure panels do not match.
- According to the data in new Figure 4G, there is an inverse correlation between GLO2 and LDHA, HK2 and GAPDH, which is significant only in the case of HK2. This needs to be corrected in the text. Visually, GAPDH correlation with GLO2 might depend on the normoxic or hypoxic condition. Did authors analyse this correlation altogether or separately within normoxic or hypoxic samples as well?

(Remarks on code availability)

Reviewer #2

(Remarks to the Author)

The authors have addressed the issues raised in my original review.

(Remarks on code availability)

Reviewer #3

(Remarks to the Author)

In the revised version of the manuscript, the authors have addressed my original comments with new data or suitable changes of the text.

(Remarks on code availability)

Reviewer #4

(Remarks to the Author)

In the manuscript entitled "Active Oxidative Metabolism and Impaired Glyoxalase System Under Increased Intracellular Oxidative Stress in Non-Small Cell Lung Cancer" Tomin and colleagues performed a comprehensive evaluation of the proteome and redox proteome of non-small cell lung cancer tissues in comparison to control (adjacent tissue). The redox proteomic approach based on Cys sequential blockage with heavy and light NEM provided robust data. The authors identified interesting associations between energy metabolism and redox homeostasis in tumor cells. They showed that lung tumor cells rewire their metabolism via protein level regulation and Cys oxidative modifications to increase glutathione

synthesis and suppress glycolysis to prevent the production of toxic advanced glycation end products. In summary, the manuscript is well written, contributes important results to the field of redox biology, and provides insights into the role of redox metabolism in lung cancer. The authors have addressed my previous concerns well, and I recommend publication of the manuscript.

(Remarks on code availability)

Reviewer #5

(Remarks to the Author)

(Remarks on code availability)

Point-by-point response to the reviewers' comments

First, we would like to thank all reviewers for their time and great suggestions to improve our manuscript.

REVIEWER COMMENTS

Reviewer #1 (Remarks to the Author):

Tomin et al. submitted a manuscript describing a specific cross-talk between ROS and metabolism in the context of Non-small cell lung cancer. For this they employ for the first time an original approach by comparing redox proteomics of tumour tissue and matched healthy tissue. They found changing levels of protein and oxidation of several enzymes in pathways related to glycolysis, mitochondrial metabolism, and reactive oxygen species. Among all the expression and oxidation changes in glucose metabolism, TCA cycle etc, the relative modifications on the metabolism of methylglyoxal are of particular interest.

While some of the findings are expected, like the increase oxidative stress in tumours compared to healthy tissue, the (presumably) hypoxia-related reduction of toxic byproducts of glucose metabolism (like MG or AGEs) can help tumour cells to grow in difficult conditions. Previous reports (Luengo et al, ref 57 in the manuscript) analysed the contribution of methylglyoxal pathway enzyme GLO1 in NSCLC and its possible therapeutic use. The manuscript is mostly a description of novel metabolic traits of NSCLC with potential interest for this and other cancers although the scope of the main findings is limited by the complex validation of these results in other patient cohorts of disease models. In addition, some relevant points raised up by the results were not confirmed and remain rather speculative.

Some major concerns need to be addressed before the article can be considered for publication.

Major:

1. Information on the clinical characteristics of the patients and of the samples is rather scarce in methods and results and could be relevant for the analyses and conclusion of the study. Although some disease biomarkers might not be regularly analysed in the clinical management of early-stage NSCLC, some other could be easily analysed on the resected sample

The patient information regarding cancer stage, PET scan values, smoker status and driver mutations (where available) was updated and added now to the **Suppl. Table S1**.

• **Resectable NSCLC is a clinical entity that includes different early NSCLC stages. Information of the tumour stage should be included in supp.**

See answer above.

• **As mentioned in the text, some genetic alterations impact metabolism (i.e. KRAS). Are driver mutations known?**

See answer above.

• **Stromal cells (immune and others) may rely on the same metabolic traits and oxidative stress in an inflammatory context like a tumour. Is there any data on the sample cellularity? Otherwise, tumour cell (or even immune cell) density should be measured to identify any possible bias on the bulk proteomic analyses.**

We sampled tiny pieces (1 mm³) of tissue (often barely reaching 100 µg of protein mass). These tiny pieces we carefully collected from exclusively tumor or healthy tissue. While this does not ensure homogenous cellularity, it does reduce the probability of large-scale immune cell infiltration compared to bulkier sampling. Nevertheless, also following the recommendation in your next comment, we examined our proteomics data for common markers of immune cell infiltration from the Lethiö *et al.* Nat Cancer 2021 dataset guided by the findings of Stankovic *et al.* 2019 (10.3389/fimmu.2018.03101) which suggested that in NSCLC tumors we should be mainly looking for either T cells (CD4+ or CD8+) or B cells. For a detailed overview, we compiled here the result of the manual search for markers of the aforementioned immune cell types in our dataset (**Table L1** below). Overall, **the findings do not point towards prominent immune cell deposition within the collected pieces of the tumor**. Still, a potential influence of stromal cells to the observed phenotype is possible.

Table L1. Status of immune cell markers in collected patient samples. Targets selected from Lethiö *et al.* 2021, specifically covering following modules: Natural.Killer.T, CD4+.Memory/Effector.T, CD8+.Naive.T, CD8+.Memory/Effector.T, B, CD4+.Naive.T (from their extended data Fig. 2f). * - target slightly up in tumor but does not pass valid value filter (min. 60 % valid values in at least one group)

Gene	Sign. Up in tumor	Sign. Down in tumor	Unchanged	Undetected	Detected <10% of total samples
ACAP1	x				
AOAH				x	
APOBEC3G			x		
BANK1				x	
CD2				x	
CD22				x	
CD247			x		
CD37				x	
CD3D			x		x
CD3E				x	
CD3G				x	
CD40	x*				
CD5				x	
CD6				x	
CD72				x	
CD74			x		
CD79B				x	
CD8A			x		x
CD96				x	
CLEC2B				x	
CORO1A			x		
CTSW				x	
CYTIP				x	
EMB				x	
ETS1				x	
EVI2B	x				
EVL		x			
FYN			x		
GIMAP4		x			
GIMAP7			x		
GLIPR1		x			

GSTK1			X	
GZMA			X	
GZMB			X	
GZMH			X	
GZMK		X		
GZMM		X		X
HLA-A		X		
HLA-DMB		X		
HVCN1	X			
IKZF3			X	
IKZF3			X	
IL4R			X	
IRF8			X	
ITGB2		X		
ITGB7			X	
LAPTM5			X	
LCK		X		X
LIMD2			X	
LSP1		X		
LY86			X	
MATK			X	
MS4A1		X		X
NR3C1	X			
OPTN		X		
PRF1		X		
PRKCB		X		
PRKCH			X	
PSTPIP1		X		X
PTPN22			X	
PTPRC	X*			
PTPRCAP		X		
PYHIN1			X	
RARRES3			X	
RASAL3	X			
RCSD1		X		
SELL		X		X
SEPTIN1		X		
SH2D1A			X	
SH2D2A			X	
SLA2			X	
SPOCK2			X	
STK17A			X	
TBX21			X	
TCL1A			X	
TMEM173	X			
TRAT1			X	
TUBA4A		X		
ZFP36L2			X	

2. Absence of similar approaches prevents validation of the oxidation data on independent cohorts. However, protein levels of enzymes could be in other NSCLC patient cohorts subjected to proteomic analyses. Lethiö et al (Nat Cancer 2021) published MS-based proteomic data, together with lots of other clinically relevant annotations, coming from NSCLC samples (mostly resectable early-stage). Since all the data comes from a single series of samples, relevance, impact and robustness would significantly improve if some results can be further validated beyond the associations between GLO2 and HIF1a (very valuable and necessary though).

We used the data from Lethiö *et al.* Nat Cancer 2021 to check for correlation of hypoxic markers (namely HK2, LDHA and GAPDH) with GLO1 and GLO2 (Table L2). Furthermore, since most of our observations were made relative to the healthy tissue, we also searched for other published proteomics datasets with matched tumor and healthy tissue. To that end, we additionally compared our data to the two following studies: lung adenocarcinoma (LUAD) dataset of 101 matched tumor and healthy tissue samples (Gillette *et al.* 2020 (doi: 10.1016/j.cell.2020.06.013)) and lung squamous cell carcinoma (LUSC) dataset of 99 matched tissue samples by Satpathy *et al.* 2021 (doi: 10.1016/j.cell.2021.07.016). In all three datasets, we tested the correlation of hypoxic genes with GLO1, GLO2 and LDHD abundance and summarized the resulting Spearman rho values in Table L2 below, marking significant correlations in bold. **For most hypoxic targets there was a negative correlation with GLO2 in at least one dataset. Also, often GLO1 was affected in the same direction as GLO2.** This will be additionally addressed in the following answers.

Table L2. Spearman rho values from individual correlation analysis in the tumor samples from three different published NSCLC proteomics studies. Significant correlations are marked in bold (Student t-test p-value < 0.05). For correlations with a p-value greater than 0.05 but smaller than 0.1, r-value is marked in italic.

Gene name	Lethiö et al. Nat Cancer. 2021		Gillette et al. Cell. 2020		Satpathy et al. Cell. 2021	
	GLO1	GLO2	GLO1	GLO2	GLO1	GLO2
GLO1	1	0.17	1	0.11	1	0.34
GLO2	0.17	1	0.11	1	0.34	1
LDHD	0.09	0.24	-0.01	0.33	0.003	-0.03
HK2	-0.20	-0.45	-0.25	-0.41	-0.11	-0.08
LDHA	0.05	-0.11	-0.19	-0.19	-0.18	-0.05
GAPDH	-0.16	-0.34	-0.06	0.12	0.009	-0.03

We incorporated data from the latter two studies in the new Fig.4D showing **prominent upregulation of hypoxic markers in tumor tissue, while GLO1, GLO2 and LDHD were downregulated** (New Fig. 4E). In addition, the two studies reported the **same elevated TCA protein signature** we observe in our tumor data (New Suppl. Fig. S2D).

3. This association between hypoxia and reduced GLO2 is mentioned in the conclusions but remains speculative. This conclusion would require to be solidified with some more experimental data (even in vitro) by inducing hypoxia and studying the expression of these enzymes. Additional data that might support this could be produced by analysing HIF1a direct targets like LDHA or HK2 (increased in tumours compared to healthy tissue). These can be discussed as surrogate markers of HIF1a activation in your same experimental samples. Same happens with the MG, which is suggested to be either less formed or more secreted. This could be experimentally addressed to be part of the conclusions.

To address hypoxia concerns raised by most reviewers, we cultured H358 cells for either 48 or 72 hours in hypoxia where we found that the rise of hypoxic markers (LDHA, HK2) was significantly inversely correlated with the abundance of GLO2 (new **Fig. 4G**). S-Lactoyl glutathione (sLG) levels were also increased by hypoxia, pointing out that acute hypoxia definitely had an impact on MG formation and regulation of the GLO pathway (new **Fig. 4F**). Of note, GLO1 levels did not correlate with hypoxic markers in our in vitro experiments (new **Fig 4H**) and GLO1 oxidation was unaffected by hypoxia (data not shown).

In our patient cohort the picture was slightly different. The rise of metabolic hypoxic markers (such as LDHA and GAPDH protein levels) was inversely correlated with MG-H1 protein modification frequency, sLG levels, and protein abundance of GLO1 but not GLO2 (new **Fig. 5D** for GAPDH; new Suppl. **Fig. S3** for LDHA) although we found only GLO2 downregulation to be significant in the group-wise comparison (**Fig. 4A**). This suggests that some of the hypoxia-sensitive enzymes affect MG formation and to an extent thereby regulate GLO1/2 activity and abundance. Our top “suspect” became GAPDH, a HIF1a target which can directly affect MG formation. High GAPDH activity can push the glycolytic flux towards pyruvate and therefore reduce the pool of available G3P/DHAP (schematic in **Fig. 5C**). GAPDH protein levels positively correlated with tumor glucose uptake (measured as 18F-fluorodeoxyglucose uptake by PET and represented in SUVmax values; new **Fig. 5F**) in our data. In TCGA data, GAPDH gene expression correlates with poorer survival of cancer patients of various etiologies (new **Fig. 5E**). Since GAPDH strongly negatively correlated with MG-H1 protein modification frequency, sLG tumor/healthy ratio and GLO1 protein levels (**Fig. 5D**) as well as GLO1 oxidation in tumors (new **Fig. 5G**), we wondered if reducing GAPDH expression/activity affects the flux towards MG formation by increasing G3P/DHAP levels creating a greater demand for GLO1-based detoxification of MG (new **Fig. 6A**).

To that end, we carried out 1) siRNA mediated knock-down of GAPDH (new **Fig. 6B,C**); or 2) pharmacological inhibition using konigic acid (KA) in A549 and H358 cells (new **Fig. 6D,E,F**), before harvesting the cells for our one-pot redox-metabolite and protein analysis. Reducing the levels of available GAPDH prominently increased GLO1 protein abundance (more so in A549 cells with a more efficient knock-down; **Fig. 6B,C**). Similarly, inhibition of GAPDH activity with KA also increased levels of GLO1 and/or GLO2 and in addition TPI (triosephosphate isomerase) (**Fig. 6F**; Suppl. **Table S8**). Of note, we first tested the ability of KA to inhibit GAPDH and opted for a concentration of 10 μ M which led to complete abolishment of GAPDH activity (**Fig. 6D**) with no additive toxicity compared to 5 μ M KA (**Fig. 6E**). In addition, both GAPDH KD and inhibition by KA increased the levels of sLG (new **Fig. 6G**).

Minor:

a) Abstract could make more emphasis on the relevance of the study in the context of cancer/NSCLC and the problematic trying to get solved.

We rephrased the abstract accordingly.

b) For a better understanding of non-experts in oxidation and metabolism, how representative is thiol oxidation of overall oxidation?

Small molecular thiols are very sensitive, with glutathione ratio being affected within minutes of oxygen exposure and improper sample handling (See Fig. 4 in <https://pubmed.ncbi.nlm.nih.gov/32079090/>). For protein thiols to be prominently affected, either longer or oxidatively stronger conditions are needed

For example, in H358 cells exposed to 100 μ M H₂O₂ in absence of serum for 1 h one can clearly see a higher protein oxidation in the H₂O₂ treatment group (see **Figure L1** below).

Figure L1: One-hour treatment with 100 μ M of H₂O₂ in serum free medium of H358 cells leads to a shift in peptide oxidation. Left: snake plot depicting distribution of peptide cysteine oxidation in control compared to H₂O₂ treated samples. Right: volcano plot of Student t-test analysis of cysteine redox affected peptides in control versus H₂O₂ treated samples. Significant hits are colored in pink (more oxidized in H₂O₂) or green (more oxidized in control; p-value < 0.05). Depicted peptides were filtered for at least three valid values in each group (N=4 per group).

Reviewer #2 (Remarks to the Author):

This manuscript is a proteomic analysis of 70 lung tumor samples and 70 matched healthy lung samples, looking at protein levels as well as levels of cysteine oxidation. In addition, the authors look at protein glycation caused by the reactive metabolite methylglyoxal. Overall, this is a very nice dataset which will certainly be useful for the community. In particular, the identification of many oxidized cysteine residues in glycolytic enzymes is exciting and could serve as a starting point for future, mechanistic studies. The current work, however, is entirely descriptive, so it could fit well as a resource paper.

As described below, I believe some of the biological conclusions / interpretations may not be warranted by the data, and would need to be changed:

1. Fig 3D- If I understand correctly, this is a correlation analysis between HIF1a mRNA levels and Glo1/2 mRNA levels, and the authors conclude this is “suggesting a strong, hypoxia driven in vivo down-regulation of GLO2 gene expression”. However, hypoxia influences HIF1a protein stability, not mRNA as far as I know? In which case this conclusion may not be warranted. I’m not sure what HIF1a mRNA levels say about HIF1a activity, since it’s mainly post-translationally regulated?

Thank you for pointing out that HIF1a and especially its mRNA levels are an unreliable readout of hypoxic state. Using other hypoxia markers, namely HK2, LDHA and GAPDH, as proxy for HIF1a activation (Fig. L2 below), we still show that most of the selected hypoxia targets are inversely correlated with GLO2 on mRNA level in LUAD, LUNG and PANCAN TCGA datasets. In addition, on protein level, both GLO1 and GLO2 were often inversely correlated with hypoxia markers across three other proteomics studies (Table L3 below) and were also found downregulated in tumor vs. healthy tissue in two independent NSCLC proteomics studies (new Fig. 4E). In our *in vitro* data, upon short term hypoxia GLO2 but not GLO1 protein levels went down with rising LDHA and HK2 protein levels (new Fig. 4G). sLG was also increased upon acute hypoxia *in vitro*, in line with less GLO2 activity (new Fig. 4F).

In our patient cohort the picture was slightly different. The rise of metabolic hypoxic markers (such as LDHA and GAPDH protein levels) was inversely correlated with MG-H1 protein modification frequency, sLG levels, and protein abundance of GLO1 but not GLO2 (new Fig. 5D for GAPDH; new Suppl. Fig. S3 for LDHA) although we found only GLO2 downregulation to be significant in the group-wise comparison (Fig. 4A). This suggests that some of the hypoxia-sensitive enzymes affect MG formation and to an extent thereby regulate GLO1/2 activity and abundance. Our top “suspect” became GAPDH, a HIF1a target which can directly affect MG formation. High GAPDH activity can push the glycolytic flux towards pyruvate and therefore reduce the pool of available G3P/DHAP (schematic in Fig. 5C). GAPDH protein levels positively correlated with tumor glucose uptake (measured as 18F-fluorodeoxyglucose uptake by PET and represented in SUVmax values; new Fig. 5F) in our data. In TCGA data, GAPDH gene expression correlates with poorer survival of cancer patients of various etiologies (new Fig. 5E). Since GAPDH strongly negatively correlated with MG-H1 protein modification frequency, sLG tumor/healthy ratio and GLO1 protein levels (Fig. 5D) as well as GLO1 oxidation in tumors (new Fig. 5G), we wondered if reducing GAPDH expression/activity affects the flux towards MG formation by increasing G3P/DHAP levels creating a greater demand for GLO1-based detoxification of MG (new Fig. 6A).

To that end, we carried out 1) siRNA mediated knock-down of GAPDH (new Fig. 6B,C); or 2) pharmacological inhibition using koniginic acid (KA) in A549 and H358 cells (new Fig. 6D,E,F), before harvesting the cells for our one-pot redox-metabolite and protein analysis. Reducing the levels of available GAPDH prominently increased GLO1 protein abundance (more so in A549 cells with a more efficient knock-down; Fig. 6B,C). Similarly, inhibition of GAPDH activity with KA also increased levels of

GLO1 and/or GLO2 and in addition TPI (triosephosphate isomerase) (Fig. 6F; Suppl. Table S8). Of note, we first tested the ability of KA to inhibit GAPDH and opted for a concentration of 10 μ M which led to complete abolishment of GAPDH activity (Fig. 6D) with no additive toxicity compared to 5 μ M KA (Fig. 6E). In addition, both GAPDH KD and inhibition by KA increased the levels of sLG (new Fig. 6G).

Figure L2. Correlation analysis between GLO2 and HIF1A target genes suggest degree of hypoxia mediated GLO2 downregulation. Hexokinase-2 (HK2), lactate dehydrogenase A (LDHA) and glyceraldehyde-3-phosphate dehydrogenase (GAPDH) gene expression is significantly inversely correlated with GLO2 in lung carcinoma (LUNG; N = 1129), lung adenocarcinoma (LUAD; N = 589). Same is true for LDHA and HK2 in all-cancer datasets (PANCAN; N = 11060).

Table L3. Spearman rho values from individual correlation analysis in the tumor samples from three different published NSCLC proteomics studies (NSCLC dataset of 141 tumors (Lehtiö *et al.* Nat Cancer 2021 (<https://doi.org/10.1038/s43018-021-00259-9>); lung adenocarcinoma (LUAD) dataset of 101 matched tumor and healthy tissue samples (Gillette *et al.* Cell 2020 (<https://doi.org/10.1016/j.cell.2020.06.013>)); and lung squamous cell carcinoma (LUSC) dataset of 99 matched tissue samples by Satpathy *et al.* Cell 2021 (<https://doi.org/10.1016/j.cell.2021.07.016>)). Significant correlations are marked in bold (Student t-test p-value < 0.05). For correlations with a p-value greater than 0.05 but smaller than 0.1, r-value is marked in italic.

Gene name	Lethiö et al. Nat Cancer. 2021		Gillette et al. Cell. 2020		Satpathy et al. Cell. 2021	
	GLO1	GLO2	GLO1	GLO2	GLO1	GLO2
GLO1	1	0.17	1	0.11	1	0.34
GLO2	0.17	1	0.11	1	0.34	1
LDHD	0.09	0.24	-0.01	0.33	0.003	-0.03
HK2	-0.20	-0.45	-0.25	-0.41	-0.11	-0.08
LDHA	0.05	-0.11	-0.19	-0.19	-0.18	-0.05
GAPDH	-0.16	-0.34	-0.06	0.12	0.009	-0.03

2. Fig 3F – if tumors have lower levels of MG-H1 and CEL, why do the authors conclude in the corresponding Results section that this “might indicate a higher presence of MG in the blood stream of lung cancer patients”? Wouldn’t it be the other way around? Alternatively, if the authors are interpreting lower MG levels in the tumor to indicate that more MG is being excreted by the tumor, this would require knowing that the rate of MG production is similar in control tissue and tumor tissue? (in particular since I don’t know of any mechanisms of active MG excretion, in which case MG excretion might not be a process that can be regulated?)

We could only speculate regarding formation vs. excretion of methylglyoxal to explain the reduced levels of MG induced protein modifications in tumors. Now, after carrying out revision experiments described in our answer above, we conclude that indeed less MG is formed in tumors with high GAPDH activity and therefore the GLO system is not induced. We changed the manuscript accordingly.

3. Line 468 “In line with ... a potential increase in MG excretion by the tumor, we observe a stark upregulation of RAGE”

Besides the comment #2 above, isn’t RAGE activation also post-translational? Would elevated levels of AGEs lead to elevated levels of RAGE protein? If not, it’s not clear that elevated levels of RAGE protein indicate elevated AGE levels. Also, where does one see elevated RAGE protein levels in the figure?

Since we observed a strong increase of RAGE in healthy tissue (or depletion of it in tumor) on protein level (volcano plot, **Suppl. Figure S2C** (RAGE gene name: AGER)), we speculated that excretion of MG by the tumor might contribute to its regulation. To test that hypothesis (and answer your question), we incubated a lung cancer cell line (A549) and non-cancerous, lung fibroblasts cells (MRC5 cell line) with rising amounts of MG (0.2, 0.5 and 1 mM) and saw absolutely no effect on RAGE protein levels in neither of the two cell lines (see **Fig. L3** below, left). However, it is perhaps noteworthy to mention that while viability of A549 cells was not affected, MRC5 cells started dying at 0.5 mM and were entirely dead upon 1 mM MG treatment (therefore also 1mM concentration was not harvested for the western blot; **Fig. L3**, right). Loss of RAGE seems thus to be rather connected to changes in tumor cellularity and its adhesion to basal membrane (<https://doi.org/10.1515/cclm-2013-0578>), and not so much to local fluctuations of AGEs. We changed the manuscript accordingly.

Figure L3: Incubation of lung cancer cells (A549) and lung fibroblasts (MRC5 cells) with rising amounts of methylglyoxal (MG; 0.2, 0.5 and 1 mM) does not affect RAGE protein levels. Left: Western blot. Right: MG concentration of 0.5 mM and higher was toxic for lung fibroblasts but not for A549 cells (N=3 per condition).

4. One of the main conclusions in the abstract is that “cancer strives to maintain oxidative metabolism amid the rise of intracellular oxidative stress”. What is the evidence for elevated oxidative stress in the tumor samples, if the tumor samples and healthy tissue samples did not separate from each other on a PCA based on the cysteine oxidation (Suppl. Fig. 2C) and the Cysred/Cysox ratio of all samples was clustered around 1 for both healthy and tumor samples? Of the cysteine residues that showed differential oxidation (Fig. 1A), there’s roughly an equal number that go up or down in the tumor samples compared to healthy tissue. Wouldn’t this suggest specific regulation rather than a global change in oxidative stress? One could argue that if tumor samples have equal levels of global cysteine oxidation compared to healthy tissue, but elevated levels of antioxidant proteins, then they have a combination of elevated oxidative stress and elevated antioxidant activity, resulting in equal levels of cysteine oxidation. But instead, tumor tissue has lower levels of antioxidative proteins compared to healthy tissue (Fig. S1E). So, this would rather speak against elevated oxidative stress in the tumor samples. So overall, this central claim does not seem to me to be supported by the data.

Thank you for raising this important criticism. In addition, during the revision process we noticed an oversight in our glutathione analysis which, after its correction, pointed out that tumors significantly accumulate glutathione. Next to examining the glutathione synthesis enzymatic machinery (which was more abundant in tumor on protein level; see new Fig. 3A,B) we also inspected glutathione’s precursors and degradation products and found them all to be prominently higher in tumor (new Fig. 3C). This rather suggests that tumors are actually protected from oxidative stress by accumulating glutathione, at least in this early tumor stage. The pool of available glutathione might allow for lower presence of antioxidative enzymes in the tumor (except –for mitochondrial SOD2 which was prominently higher in tumor). We changed the manuscript accordingly.

Minor Issues:

1. Line 229 “while almost three times more significantly more oxidized proteins”

Extra “more”

Corrected.

2. Fig 1E – what does “Genes (%)” mean? I assume that’s the percent of genes in that category that are in the gene set? Please specify.

The figure should also show FDR or corrected p-values on the bars.

FDR values were added and Genes (%) explained in the figure legend of now Fig. 1D.

3. Line 379 “we addressed levels of sLG,”

We assessed levels ?

Corrected.

Reviewer #3 (Remarks to the Author):

Oxidative modification of cysteines in proteins has emerged as a significant cell regulatory mechanism. Cancer cells use multiple mechanisms to survive through oxidative stress, in part through oxidation-mediated modification of protein activity. Therefore, identification of oxidised proteins is an important step for understating mechanisms that drive cancer cell survival. Several studies have reported methods to identify oxidatively modified proteins in cultured cells, but it has been unclear whether the same proteins, and to which extent, are oxidatively modified in tumour tissues. In this manuscript, Tomin et al. address this point by using a method they previously developed to identify proteins that carry oxidatively modified cysteines in normal and tumour tissue from human non-small cell lung cancer (NSCLC) patients.

The authors show that approx. 10%-30% of the quantified oxidised cysteines are differentially oxidised in tumour vs. paired non-tumour tissue, and several of the proteins carrying these cysteines can be linked to major oncogenic pathways. Furthermore, the authors provide evidence that oxidation in several proteins involved in central carbon metabolism is higher in tumours compared to non-tumour tissue. Tomin et al. also provide evidence that proteins of the glyoxylate pathway are significantly enriched among those that are more oxidised between tumour and non-tumour tissues. The glyoxylate pathway breaks down methylglyoxal, a toxic metabolic byproduct of central carbon metabolism. Pivoting on this observation, the authors provide evidence that a combination of cysteine oxidation and changes in the abundance of glyoxylate pathway enzymes may lead to increased methylglyoxal levels in tumours. However, methylglyoxal adducts are higher in surrounding tissue compared to tumours, suggesting that tumours deal with glyoxylate system inactivation by increased excretion of methylglyoxal.

Overall, while evidence of causal links between the various observations the authors describe is missing, the main strength of this study is the novelty of analysing protein oxidation status in human tumours. The following points may help strengthen the author's claims.

1. Lines 385-405 and discussion lines 506-524: The authors' observation that low GLO2 levels correlate with poorer patient survival supports the idea that loss of glyoxylase system activity promotes tumour growth, however it also raises the possibility that suppressed GLO2 protein expression may suffice to deactivate this pathway. Given that the backbone of this study is the identification of oxidised proteins in patient samples, further investigation of the context in which GLO1 oxidation may be relevant is needed. To address this issue, the authors could consider further analyses of their dataset:

- test whether GLO1 oxidation is more likely to occur in tumours with high GLO2 levels (thereby showing that GLO1 oxidation is an alternative route for inactivating the pathway when GLO2 is present).

Following your suggestion, we tested the correlation between GLO2 abundance and GLO1 Cys^{red}/Cys^{ox} of Cys¹³⁹ (in short: GLO1 L/H) but could not observe any significant trends (see Fig. L4 below (also new Fig 4B); right panel). One reason could be that, contrary to GLO2, GLO1 protein expression is significantly inversely correlated with GLO1 L/H (Fig. L4, left graph, Fig. 4B) suggesting that when more GLO1 is oxidized more GLO1 is produced to compensate for loss of function by oxidation.

Figure L4: GLO1 but not GLO2 protein levels significantly correlate with GLO1 Cys^{red}/Cys^{ox} of Cys¹³⁹ oxidation status. Number of tested pairs: 58-59. Of note, lower L/H ratios means more oxidation.

- Furthermore, to strengthen the proposed link between the glyoxylase enzyme oxidation status and hypoxia, can the authors assess how well hypoxic markers in the 70 tumours they analysed correlate with GLO1 oxidation?

Following up on your second remark, we tested the correlation of a few hypoxic markers (namely HK2, LDHA, GAPDH) with GLO1 L/H (see Fig. L5 below). In addition, we observed that LDHA and GAPDH were inversely correlated with sLG levels, the product of GLO1 activity (new Fig. 5D (GAPDH) and new Suppl. Fig. S3 (LDHA)).

Figure L5: GAPDH (but not LDHA nor HK2) protein abundance is significantly inversely correlated with GLO1 Cys¹³⁹ Cys^{red}/Cys^{ox} levels (i.e., GAPDH is directly correlated with more GLO1 oxidation). Number of tested pairs: 59.

GAPDH being significantly inversely correlated with GLO L/H was an additional reason for us to suspect that GAPDH might act as a hypoxia-dependent regulator of the GLO system activity.

High GAPDH activity can push the glycolytic flux towards pyruvate and therefore reduce the pool of available G3P/DHAP (schematic in Fig. 5C). GAPDH protein levels positively correlated with tumor glucose uptake (measured as 18F-fluorodeoxyglucose uptake by PET and represented as SUVmax values; new Fig. 5F) in our data. In TCGA data, GAPDH gene expression correlated with poorer survival (new Fig. 5E). Since GAPDH strongly negatively correlated with MG-H1 protein modification frequency, sLG tumor/healthy ratio and GLO1 protein levels (new Fig. 5D) as well as GLO1 oxidation in tumors (new Fig. 5G), we wondered if reducing GAPDH expression/activity affects the flux towards MG

formation by increasing G3P/DHAP levels creating a greater demand for GLO1-based detoxification of MG (new **Fig. 6A**).

To that end, we carried out 1) siRNA mediated knock-down of GAPDH (new **Fig. 6B,C**); or 2) pharmacological inhibition using konigic acid (KA) in A549 and H358 cells (new **Fig. 6D,E,F**), before harvesting the cells for our one-pot redox-metabolite and protein analysis. Reducing the levels of available GAPDH prominently increased GLO1 protein abundance (more so in A549 cells with a more efficient knock-down; **Fig. 6B,C**). Similarly, inhibition of GAPDH activity with KA also increased levels of GLO1 and/or GLO2 and in addition TPI (triosephosphate isomerase) (**Fig. 6F**). Of note, we first tested the ability of KA to inhibit GAPDH and opted for a concentration of 10 μ M which led to complete abolishment of GAPDH activity (**Fig. 6D**) with no additive toxicity compared to 5 μ M KA (**Fig. 6E**). In addition, both GAPDH KD and inhibition by KA increased the levels of sLG (new **Fig. 6G**).

2. Line 406-onwards and related to the last point in #1: The authors use HIF1 α mRNA expression as an indicator of hypoxia, however, the main regulatory mechanism of hypoxia response is post-transcriptional stabilisation of HIF1 α . HIF1 α gene targets (such as CA9) would be more suitable for inferring hypoxia and the authors should test the correlation between HIF1 α targets and GLO1/2.

Thank you for pointing out that HIF1 α and especially its mRNA levels are an unreliable readout of hypoxic state. Using other hypoxia markers, namely HK2, LDHA and GAPDH, as proxy for HIF1 α activation (**Fig. L2** below), we still show that most of the selected hypoxia targets are inversely correlated with GLO2 on mRNA level in LUAD, LUNG and PANCAN TCGA datasets. In addition, on protein level, both GLO1 and GLO2 were often inversely correlated with hypoxia markers across three other proteomics studies (**Table L3** below) and were also found downregulated in tumor vs. healthy tissue in two independent NSCLC proteomics studies (new **Fig. 4E**). In our *in vitro* data, upon short term hypoxia GLO2 but not GLO1 protein levels went down with rising LDHA and HK2 protein levels (new **Fig. 4G**). sLG was also increased upon acute hypoxia *in vitro*, in line with less GLO2 activity (new **Fig. 4F**).

In our patient cohort the picture was slightly different. The rise of metabolic hypoxic markers (such as LDHA and GAPDH protein levels) was inversely correlated with MG-H1 protein modification frequency, sLG levels, and protein abundance of GLO1 but not GLO2 (new **Fig. 5D** for GAPDH; new **Suppl. Fig. S3** for LDHA) although we found only GLO2 downregulation to be significant in the group-wise comparison (**Fig. 4A**). This suggests that some of the hypoxia-sensitive enzymes affect MG formation and to an extent thereby regulate GLO1/2 activity and abundance. Our top "suspect" became GAPDH, a HIF1 α target which can directly affect MG formation as described in our answer to your previous comment.

Figure L2. Correlation analysis between GLO2 and HIF1A target genes suggest degree of hypoxia mediated GLO2 downregulation. Hexokinase-2 (HK2), lactate dehydrogenase A (LDHA) and glyceraldehyde-3-phosphate dehydrogenase (GAPDH) expression in significantly inversely correlated with GLO2 in lung carcinoma (LUNG; N = 1129), lung adenocarcinoma (LUAD; N = 589). Same is true for LDHA and HK2 in all-cancer datasets (PANCAN; N = 11060).

Table L3. Spearman rho values from individual correlation analysis in the tumor samples from three different published NSCLC proteomics studies (NSCLC dataset of 141 tumors (Lehtiö *et al.* Nat Cancer 2021 (<https://doi.org/10.1038/s43018-021-00259-9>); lung adenocarcinoma (LUAD) dataset of 101 matched tumor and healthy tissue samples (Gillette *et al.* Cell 2020 (<https://doi.org/10.1016/j.cell.2020.06.013>)); and lung squamous cell carcinoma (LUSC) dataset of 99 matched tissue samples by Satpathy *et al.* Cell 2021 (<https://doi.org/10.1016/j.cell.2021.07.016>)). Significant correlations are marked in bold (Student t-test p-value < 0.05). For correlations with a p-value greater than 0.05 but smaller than 0.1, r-value is marked in italic.

Gene name	Lehtiö et al. Nat Cancer. 2021		Gillette et al. Cell. 2020		Satpathy et al. Cell. 2021	
	GLO1	GLO2	GLO1	GLO2	GLO1	GLO2
GLO1	1	0.17	1	0.11	1	0.34
GLO2	0.17	1	0.11	1	0.34	1
LDHD	0.09	0.24	-0.01	0.33	0.003	-0.03
HK2	-0.20	-0.45	-0.25	-0.41	-0.11	-0.08
LDHA	0.05	-0.11	-0.19	-0.19	-0.18	-0.05
GAPDH	-0.16	-0.34	-0.06	0.12	0.009	-0.03

3. In Fig. 1B, the authors calculate enrichments of the identified oxidised proteins according to their cellular location; they then use these enrichments to argue that the location of oxidised proteins differs between tumour (T) and non-tumour (NT) tissues. However, changes in the T vs NT proteome

are likely to affect (and, indeed, may account for) these enrichments. Apologies if I missed this, but I did not find information indicating that the authors took into account differences in the background proteomes of T and NT for these enrichment calculations, which is essential in order to provide meaningful enrichment statistics.

Thank you for this valuable remark. The background was in all cases the whole human proteome. To avoid any potential bias introduced this way, we decided to omit GOCC enrichment and rather simply analyze the subcellular localization (using information provided by the UniProt database) of the uniquely more oxidized proteins in tumor or healthy tissue (see new **Fig. 1B**). For GOBP analysis, as there all significantly redox affected proteins were used as input (removing the group specific bias), we kept the whole human proteome as the background.

4. Line 299: the authors attribute lack of GSH/GSSG ratio differences to increased GSH synthesis. This idea would be better supported by calculating the sum of GSH and GSSG abundances (as an indicator of the total glutathione pool size), which, given the increased GSSG, is likely to be found increased, in support of the authors' claim. Presenting these data is important, as, increased GSH synthesis enzyme gene expression (shown in Fig. S3B) correlates with but does not prove increased GSH synthesis. Furthermore, given that the NRF2 pathway is important in tumour antioxidant responses and may drive glutathione synthesis, the authors may want to investigate whether their proteomic or redox data (the NRF2 regulator Keap1 carries regulatory reactive cysteines) for evidence of NRF2 pathway activation.

Thank you for this important remark. As we returned to reanalyze the glutathione data to address your comment, we noticed an oversight in calculation of glutathione values which affected several samples. We sincerely apologize for this. The tables containing raw data exports and calculations have been also submitted as the accompanying source files to the manuscript. Upon this correction, we actually observe a significant change in GSH content of the tumor. Tumors have significantly higher levels of both GSH, GSSG and then, of course, also their sum (see new **Fig. 3B**). The GSH/GSSG ratio remained similar (data not shown). In addition, to further address glutathione metabolism, we also measured all other relevant thiol pairs which are involved in glutathione synthesis as well as degradation (new **Fig. 3C**). Next to glutathione, all of the other relevant thiols were also increased in the tumor compared to healthy tissue, especially in their reduced form.

Regarding NRF2, we could not draw a clear conclusion. Keap1 protein is indeed upregulated in tumor. All other NRF2 targets (summarized in the **Table L4** below), however, did not show a clear trend. Targets were selected according to the following publications: <https://doi.org/10.1016/j.cbi.2011.01.026>, <https://www.nature.com/articles/s41419-021-04486-x>, <https://pmc.ncbi.nlm.nih.gov/articles/PMC7298623/>. Unfortunately, the redox sensitive cysteine of Keap1 was not detected in the redox data.

Table L4: Protein abundance status of NRF2 targets in tumor vs. healthy tissue. Green represents target down-while orange represents targets up-regulated in tumor. Targets in white were unchanged.

Target	Status in tumor vs. healthy	Paired t-test significant	Status code
GCLC	Green	Yes	down
GCLM	White	N/A	unchanged
GPX2	White	N/A	unchanged
GSTA1	White	N/A	unchanged
GSTM1	Green	Yes	down
GSTM2	Green	Yes	down
GSTM3	Green	Yes	down
GSTP1	White	N/A	unchanged
NQO1	Orange	Yes	up
TXN	White	N/A	unchanged
TXNRD1	White	N/A	unchanged
SRXN1	Orange	Yes	up
G6PD	White	N/A	unchanged
PGD	Orange	Yes	up
IDH1	Orange	Yes	up
ME1	White	N/A	unchanged
HMOX1	Green	Yes	down
FTH1	Orange	Yes	up
AKR1B10	Orange	Yes	up
SOD2	Orange	Yes	up
GSS	Orange	Yes	up
CAT	Green	Yes	down
SOD1	White	N/A	unchanged
KEAP1	Orange	Yes	up
CUL3	Green	Yes	down

5. Furthermore, while glycolytic intermediate re-routing, as the authors propose, may contribute to the increased glutathione reduction to produce GSSG, increased glutathione synthesis per se could also be explained by divergence of glucose-derived carbons into serine/glycine synthesis. The authors should discuss this possibility.

This is an excellent point. To answer your question, we checked the abundance of proteins involved in *de novo* serine/glycine synthesis (Yang and Vousden, 2016; <https://doi.org/10.1038/nrc.2016.81>) namely PHGDH, PSAT1, PSPH, SHMT1 and 2 and observed that most of them (with exception of PHGDH and SHMT1) were all upregulated in cancer. The data was now visually merged with the glutathione synthesis pathway in the new Fig. 3A. Together with the increase in the thiol building blocks of glutathione, this strengthens our conclusion that more glutathione synthesis is carried out by the tumors. We changed the manuscript accordingly.

Other points:

- Line 462: define "AGEs" acronym.

Done.

- Fig. 3A: sLG graph y axis "pmol/mg": does the mass unit refer to tissue protein or dry/wet tissue weight?

For all small-molecular thiols presented, it refers to the protein content. This information was now added to either axes titles or figure legends.

Reviewer #4 (Remarks to the Author):

In the manuscript entitled “Active Oxidative Metabolism and Impaired Glyoxalase System Under Increased Intracellular Oxidative Stress in Non-Small Cell Lung Cancer” Tomin and colleagues performed comprehensive evaluation of proteome and redox proteome of non-small cell lung cancer in comparison to control (adjacent tissue). The authors purpose the impact of protein redox modulation in metabolic rewiring of tumor cells, especially of glyoxalase system. Complementary analyses such as estimation of glutathione content and database search were done. In conclusion authors report that both redox and protein abundance changes contribute to the rise of intracellular oxidative stress and deactivate glyoxalase system which is compensated by higher excretion or lower production of methylglyoxal.

In summary, the manuscript is well written, contributes important results to the field of redox biology and provides novel insights into the role of redox metabolism in lung cancer. However, the manuscript contains several uncertainties, the clarification of which would contribute significantly to the strength of the claims made and to the reproduction of the methods.

Major comments

1. The healthy tissues were dissected from the different part of the lung of the same patient. Authors should demonstrate the definition/picture/cytology of non-cancerous changes in the healthy tissue. Are the ratios of protein abundances between healthy and tumor tissues done within the same patient?

Yes, healthy and tumor tissue was always collected from the same patient (from total of 70 patients). While we do not demonstrate cytological images, the non-cancerous tissue was harvested at least 3 cm away from the tumor and this selection was guided by the pathologist present on site. This is now clearly described in the manuscript.

2. It is surprising that the ratio between oxidized and reduced peptides is 1 in both tissues. According to the previous publications the median peptide oxidation is ranging between 5-20%. Can authors explain such discrepancy?

In our hands, 30-50 % oxidation is the value we get out from our redox metabolite and protein one-pot approach and might be connected with the fact that unlike studies mentioned, we:

1) do not carry out any enrichment steps nor use any detergents during the sample preparation. In this regard, we observe similar median oxidation ratios as another none-enrichment approach - the SICyLIA method which reported median log₂ ratio of 0 for kidney cells (van der Reest et al., *Nat Comm*, 2018. <https://www.nature.com/articles/s41467-018-04003-3>). However, this median ratio can certainly be strongly shifted by oxidative stress. For example, in H358 cells exposed to 1 h 100 μM H₂O₂ in absence of serum one can clearly see a higher protein oxidation in the H₂O₂ treatment group (see **Fig. L1** below).

2) are working with cancerous cells and tissues. A recent redox proteomic study by Tan *et al. J Trans Med*, 2024 (<https://doi.org/10.1186/s12967-024-05068-z>) also reported median oxidation ratios spanning from 0.8 to 1.2 for pancreatic cancer and non-cancerous transformed pancreatic cells.

Generally speaking, with cysteine redox proteomics being still an underrepresented PTM omics analysis, it is hard to conduct an exact cross-study comparison. Another example of a prominent discrepancy would be the very recent PACREDOX study (still only a preprint; <https://www.biorxiv.org/content/10.1101/2024.12.21.629874v1.full#F5>) which reports ~30% median Cys oxidation for myocardium of pigs, while the OXIMOUSE study puts heart at the average of 10% oxidation (<https://doi.org/10.1016/j.cell.2020.02.012>).

In conclusion, we agree with the reviewer that there are discrepancies in reported cysteine redox studies, which actually points out the need for more redox studies, especially across different pathologies, to be able to draw clear conclusions.

Figure L1: One-hour treatment with 100 μ M of H₂O₂ in serum free medium leads to a shift in peptide oxidation. Left: snake plot depicting distribution of peptide cysteine oxidation in control compared to H₂O₂ treated samples. Right: volcano plot of Student t-test analysis of cysteine redox affected peptides in control versus H₂O₂ treated samples. Significant hits are colored in pink (more oxidized in H₂O₂) or green (more oxidized in control; p-value < 0.05). Depicted peptides were filtered for at least three valid values in each group (N=4 per group).

3. In Fig. 1B authors showed that tumor tissues in contrast to healthy tissue “have relatively more oxidation-affected intracellular proteins”. The analysis involved both significantly reduced and oxidized peptides. It would be more correct to use the term “redox changed” peptides. Separate GO analysis for the reduced and oxidized peptides would clarify the demonstration.

Thank you for this remark. As a similar point was raised by another reviewer, we now omitted GOCC enrichment for clarity and only commented on the subcellular distribution of proteins uniquely oxidized in either tumor or healthy tissue annotated by information from Uniprot (see new Fig. 1B).

4. In line 284 authors claim that they showed for the first-time large number of glucose-related metabolic enzymes are susceptible to redox modulation in vivo, please see the papers of Xiao et al. (PMC8164166) or Pimkova et al. (PMC9157258).

Thank you for this remark. We corrected the manuscript to stress we refer to novelty in cancer patients' *in vivo* findings.

5. In Figure 2, some of the enzymes such as HK2 is demonstrated as the one with unaffected redox state, but searching in Table S3 it was not identified at all. It would be more accurate to distinguish between these findings.

Thank you for your valuable remark. We now added additional marking for enzymes which were actually not found in the redox dataset in Fig. 2.

6. Line 379, to be able to claim if the activity of GLO1 is lower both the substrate (MG) and the product should be determined.

Methylglyoxal is a rather challenging analyte to measure from long-term stored samples due to its high chemical reactivity. As the samples for this study were collected over the period of around one year, by the time we processed the samples, we expected most of the free MG to have already reacted. That

is why we opted to use MG-H1 and CEL, the MG-derived protein modifications, as a proxy for substrate (MG) content.

7. Were GLO1 or GLO2 shown as targets of HIF1A according to available databases (e.g. <https://guolab.wchscu.cn/hTFtarget/#!/>)

Thank you very much for this suggestion! Indeed, GLO2 is a more likely HIF1A target than GLO1 in the suggested database. This is the result for **GLO1** around TFs whose symbol starts with “H”:

No. of datasets	TF	Tissue	No. of peaks (total/average)	No. of peaks in gene body (total/average)	No. of peaks around TSS (total/average)
5	GATA1	bone marrow	21/4	13/2	8/1
1	GATA1	peripheral blood	1/1	1/1	0/0
1	GATA4	embryo	2/2	1/1	1/1
1	GATA4	skin	3/3	2/2	1/1
1	GLI2	colon	2/2	1/1	1/1
1	GMEB2	colon	2/2	1/1	1/1
1	HCFC1	bone marrow	2/2	1/1	1/1
2	HCFC1	cervix	5/2	2/1	3/1
1	HCFC1	kidney	2/2	1/1	1/1
1	HDAC2	blood	2/2	1/1	1/1
1	HDAC2	prostate	2/2	1/1	1/1
1	HDAC6	other	2/2	1/1	1/1
1	HEY1	bone marrow	2/2	1/1	1/1
1	HNF4A	other	3/3	2/2	1/1
1	HOXA6	colon	2/2	1/1	1/1
1	IRF1	bone marrow	2/2	1/1	1/1
1	IRF1	Pancreatic Ductal	4/4	2/2	2/2

And this is how it looks for GLO2:

No. of datasets	TF	Tissue	No. of peaks (total/average)	No. of peaks in gene body (total/average)	No. of peaks around TSS (total/average)
1	HCFC1	cervix	2/2	1/1	1/1
1	HDAC1	blood	2/2	1/1	1/1
1	HDAC1	bone marrow	5/5	3/3	2/2
1	HDAC2	blood	3/3	2/2	1/1
1	HDAC2	breast	4/4	3/3	1/1
3	HDAC2	embryo	9/3	7/2	2/0
1	HDAC2	liver	10/10	5/5	5/5
1	HDAC2	prostate	2/2	1/1	1/1
1	HEY1	bone marrow	3/3	2/2	1/1
1	HIF1A	blood	1/1	0/0	1/1
1	HIF1A	bone	4/4	2/2	2/2
1	HIF1A	breast	4/4	3/3	1/1
1	HIF1A	other	1/1	0/0	1/1
1	HIF1A	patient	1/1	0/0	1/1
1	HIF1A	renal tubular cells	1/1	0/0	1/1
1	HINFP	colon	2/2	1/1	1/1
2	HNF4A	colon	4/2	1/0	3/1
1	HNF4A	intestinal	3/3	1/1	2/2
5	HNF4A	liver	18/3	8/1	10/2
4	HNF4A	other	21/5	11/2	10/2
1	HNF4G	liver	7/7	2/2	5/5
1	HOXA1	colon	2/2	1/1	1/1
1	HOXA6	colon	2/2	1/1	1/1
1	IRF1	bone marrow	2/2	1/1	1/1

8. In TCGA analysis, the GLO1 and GLO2 expression is evaluated in three types of lung tissues: healthy, tumor-adjacent, tumor. Which of them are relevant to the analyses done by authors? For the simplifications only the TCGA analysis of compared tissues would be good to show.

Our healthy samples are matched healthy samples but were not collected directly next to the tumor (but a minimum of 3 cm away from the tumor). Thus, they cannot be considered tumor adjacent. On the other hand, they cannot be considered entirely healthy lung either, considering there is a tumor at least 3 cm away. So, technically neither of the two TCGA healthy categories matches our sample type accurately, with our samples being perhaps more on the healthy tissue side; but we believe both categories provide valuable information with regards to gene expression of the GLO system.

9. It would be useful to show the summary of open search analysis results, such as how many proteins were identified and how many of those were modified with MG.

Oxidation and MG-derived modifications were now marked in **Suppl. Table S7** in bold letters and targeted search elaborated in more details with regards to number of MG-H1 modified peptides/corresponding proteins.

10. In Fig. 3F it seems that majority of identified peptides is MG-H1 modified. Are indeed such many peptides MG-H1 modified?

Only MG-H1 modified peptides are displayed here; to be precise, the total of 1525 peptides which were identified in at least 10 samples of at least one group (tumor or healthy). They represent less than 3% of all identified peptides.

11. Authors point to the increased glycolytic activity of lung tumor tissue; may they demonstrate it?

To address your comment, we collected PET data from the patients and observed a direct correlation of PET intake by the tumor with glycolytic markers such as GAPDH suggesting some degree of glycolytic dependence of the tumors (new Fig. 5F). Furthermore, the majority of the glycolytic enzymes were more abundant in tumor compared to healthy tissue on protein level (Fig. 2, Suppl. Table S2).

12. Authors conclude that both the redox change of GLO1 or protein abundance of GLO2 contribute to the phenotype of tumor cells. It would be very helpful if e.g. using lung cell line authors could conclude which of those two conditions contribute to the phenotype. For example, by modulation of redox environment using antioxidants or overexpression of GLO2.

Thank you for this remark. We performed additional *in vitro* experiments to investigate the influence of hypoxia and GAPDH activity on the GLO system.

Upon short term hypoxia GLO2 but not GLO1 protein levels went down in lung cancer cells with rising LDHA and HK2 protein levels (new Fig. 4G,H). sLG was also increased upon acute hypoxia *in vitro*, in line with less GLO2 activity (new Fig. 4F).

Since GAPDH strongly negatively correlated with MG-H1 protein modification frequency, sLG tumor/healthy ratio and GLO1 protein levels (new Fig. 5D) as well as GLO1 oxidation in tumors (new Fig. 5G), we wondered if reducing GAPDH expression/activity affects the flux towards MG formation by increasing G3P/DHAP levels creating a greater demand for GLO1-based detoxification of MG (new Fig. 6A). To that end, we carried out 1) siRNA mediated knock-down of GAPDH (new Fig. 6A,B,C); or 2) pharmacological inhibition using koningic acid (KA) in A549 and H358 cells (new Fig. 6A,D,E,F), before harvesting the cells for our one-pot redox-metabolite and protein analysis. Reducing the levels of available GAPDH prominently increased GLO1 protein abundance (more so in A549 cells with a more efficient knock-down; new Fig. 6B,C). Similarly, inhibition of GAPDH activity with KA also increased levels of GLO1 and/or GLO2 and in addition TPI (triosephosphate isomerase) (new Fig. 6F). Of note, we first tested the ability of KA to inhibit GAPDH and opted for a concentration of 10 μ M which led to complete abolishment of GAPDH activity (Fig. 6D) with no additive toxicity compared to 5 μ M KA (Fig. 6E). In addition, both GAPDH KD and inhibition by KA increased the levels of sLG (Fig. 6G).

We conclude that GLO1 plays a more important role in acute MG detoxification and that GLO2 is downregulated in hypoxia, probably to avoid further acidification by its product D-lactate.

Moreover, contrary to GLO2, GLO1 protein expression is significantly inversely correlated with GLO1 Cys¹³⁹ L/H (Cys^{red}/Cys^{ox}) in tumors (Fig. L4, also new Fig. 4B) suggesting that when more GLO1 is oxidized more GLO1 is produced to compensate for loss of function by oxidation.

Figure L4: GLO1 but not GLO2 protein levels significantly correlate with GLO1 Cys^{red}/Cys^{ox} of Cys¹³⁹ oxidation status. Number of tested pairs: 58-59. Of note, lower L/H ratios means more oxidation.

13. Are available data of known mutations present in analyzed tissues? This is important to exclude the possibility of enzymes deactivation by mutation.

Mutation information where available was now added to the Suppl. Table S1. Only very few patients had prominent (known) mutations confirmed. Of note, GLO2 was deactivated in virtually all samples.

Minor comments

1. Inconsistency in labeling of Supplementary figures.

2. There are no links to Supplementary figure 1A,1B 1E.

Supplement and figures were now drastically rearranged.

3. In Fig. 1A, it is more common to present ratio of oxidized vs. reduced and tumor vs. healthy. The Fig. would be easier to read when authors demonstrate e.g. that RACK1 and CAV1 are the most oxidized proteins in tumors.

This is merely a matter of taste. It is our personal preference to present the glutathione ratio as light/heavy label ratio corresponding to red/ox ratio and we use the same approach (L/H) across all our studies. We also consistently present first healthy and then tumor in all our figures.

4. RACK1 and CAV1 proteins are the most significantly changed but not the most oxidized as written in line 207.

Corrected.

5. In Fig. 1A line 244 “each dot represents an individual cysteine residue”. More correct would be individual peptide, since some of the peptides contain more than one cysteine.

Our script processes the data in such way that different peptides harboring the same cysteine residue are averaged. Therefore, saying each point represents an individual peptide would also be inaccurate. We clarified it now in the text a bit better.

6. The y axis marked as “-log FDR corrected p-value” according to supplementary table S3 rather shows the non-corrected p-values.

Axes are corrected now.

7. In Supplementary table 3, column “C” is probably not the logarithm as written in the header.

Corrected, thank you.

8. For the more fluent reading I would recommend to put the link to Fig. S2 and mention the GOBP analysis was done on protein abundances and showed relevant biological processes to be enriched in line 276.

Done.

9. In line 402 it is unclear what “usual” means.

Part of the text was removed during revision.

10. Line 462 shortage “AGE” not defined.

Part of the text was removed during revision. AGE is now defined.

Reviewer #5 (Remarks to the Author):

We would like to thank all the reviewers to their valuable comments throughout this revision and are delighted to hear that the new manuscript meets requirements for publication.

Reviewer #1 (Remarks to the Author):

Authors addressed successfully the raised questions and provided significant amount of new data that gives a better idea of a tumour metabolic rewiring to adapt glucose metabolism and oxidative stress in disease context. Now they have validated up to some extent the findings from patient samples, and open new avenues in this filed in NSCLC.

Here we provide some additional minor comments:

- **Figure 4 needs revision, since text, caption and figure panels do not match.**
- **According to the data in new Figure 4G, there is an inverse correlation between GLO2 and LDHA, HK2 and GAPDH, which is significant only in the case of HK2. This needs to be corrected in the text. Visually, GAPDH correlation with GLO2 might depend on the normoxic or hypoxic condition. Did authors analyse this correlation altogether or separately within normoxic or hypoxic samples as well?**

Thank you for your comments. Figure 4 annotation was now corrected and significance of correlation was now clearly stated. Regarding the hypoxia: GAPDH was in this context used as one of the hypoxia markers (next to LDHA and HK2). Correspondingly, both normoxic and hypoxic conditions were observed together (as hypoxia influences GAPDH abundance).